# Flipped Classroom: Effective Teaching for Chaotic Time Series Forecasting

## Abstract

Gated RNNs like LSTM and GRU are the most common choice for forecasting time series data reaching state of the art performance. Training such sequence-to-sequence RNNs models can be delicate though. While gated RNNs effectively tackle exploding and vanishing gradients, there remains the exposure bias problem provoked by training sequence-to-sequence models with teacher forcing. Exposure bias is a concern in natural language processing (NLP) as well and there are already plenty of studies that propose solutions, the most prominent probably being scheduled sampling. For time series forecasting, though, the most frequent suggestion is training the model in free running mode to stabilize its prediction capabilities over longer horizons. In this paper, we demonstrate that exposure bias is a serious problem even or especially outside of NLP and that training such models free running is only sometimes successful. To fill the gap, we are formalizing curriculum learning (CL) strategies along the training as well as the training iteration scale, we propose several completely new curricula, and systematically evaluate their performance in two experimental sets. We utilize six prominent chaotic dynamical systems for these experiments. We found that the newly proposed increasing training scale curricula with a probabilistic iteration scale curriculum consistently outperforms previous training strategies yielding an NRMSE improvement up to 81% over free running or teacher forced training. For some datasets we additionally observe a reduced number of training iterations and all models trained with the new curricula yield higher prediction stability allowing for longer prediction horizons.

## 1 Introduction

[1] Advanced Recurrent Neural Networks (RNNs) such as Long Short Term Memory (LSTM) (Hochreiter & Schmidhuber, 1997) and Gated Recurrent Unit (GRU) (Cho et al., 2014) achieved significant results in predicting sequential data (Chung et al., 2014; Nowak et al., 2017; Yin et al., 2017). Such sequential data can for example be textual data as processed for Natural Language Processing (NLP) tasks where RNN models were the method of choice for a long time, before feed-forward architectures like transformers showed superior results in processing natural language data (Devlin et al., 2018; Yang et al., 2019; Radford et al., 2019; Brown et al., 2020). Shifting the view to the field of modeling dynamical or even chaotic systems, encoder-decoder RNNs are still the method of choice for forecasting such continuous time series data (Wang et al., 2019; Thavarajah et al., 2021; Vlachas et al., 2018; Sehovac et al., 2019; Sehovac & Grolinger, 2020; Shen et al., 2020).

Nevertheless encoder-decoder RNNs do have a flaw as is well known from the time when they were the type of model to use for NLP applications. This shortcoming is termed *exposure bias* that can appear when teacher forcing is used for training the model of question. Teacher forcing is the strategy that is typically applied when training RNNs for time series sequence-to-sequence tasks (Williams & Zipser, 1989) regardless of the type of data. To understand the problem we first give a quick side note about the motivation behind teacher forcing. Its main advantage is that it can significantly reduce the number of steps a model needs to

---

[1]To the reviewers: Upon deanonymization our code for the experiments will be made available as a replication package on Github. The datasets used in this paper were already published on Dataverse.

converge during training and improve its stability (Miao et al., 2020). However, teacher forcing may result in worse model generalization due to a discrepancy between training and testing data distribution (*exposure bias*). It is less resilient against self induced perturbations caused by prediction errors in the inference phase (Sangiorgio & Dercole, 2020).

Several authors propose methods to mitigate the *exposure bias* and reduce its negative effect with the help of training strategies (Bengio et al., 2015; Nicolai & Silfverberg, 2020; Lamb et al., 2016; Liu et al., 2020; Dou et al., 2019). However, all these methods address NLP tasks, such as, text translation or image captioning. We want to focus on *exposure bias* mitigation while forecasting chaotic systems. Those systems directly depend on their past but also tend to be easily irritated by perturbations. A model needs to be especially resilient against small perturbations when auto-regressively predicting future values, otherwise those small errors will quickly accumulate to larger errors (Sangiorgio & Dercole, 2020). We argue that sequence-to-sequence models predicting chaotic time series are even more vulnerable to *exposure bias* and will thus more likely fail to forecast larger horizons.

Besides *exposure bias* mitigation, the field of forecasting and analyzing (chaotic) dynamical systems has intensively been studied with many RNN-based approaches proposed to stabilize the training process and preventing exploding gradients. Most of these studies propose architectural tweaks or even new RNN architectures considering the specifics of dynamical systems and their theory (Lusch et al., 2018; Vlachas et al., 2018; Schmidt et al., 2019; Champion et al., 2019; Chang et al., 2019; Rusch & Mishra, 2020; Erichson et al., 2020; Rusch et al., 2021; Li et al., 2021).

Monfared et al. (2021) performed a theoretical analysis relating RNN dynamics to loss gradients and argue that this analysis is especially insightful for chaotic systems. With this in mind they suggest a kind of sparse teacher forcing (STF) inspired by the work of Williams & Zipser (1989) that uses information about the degree of chaos of the treated dynamical system. As a result, they form a training strategy that is applicable without any architectural adaptations and without further hyperparameters. Their results using a vanilla RNN, a piecewise linear recurrent neural network (PLRNN) and an LSTM for the Lorenz (Lorenz, 1963) and the Rössler (Rössler, 1976) systems show clear superiority of applying chaos dependent STF.

Reservoir computing RNNs were successfully applied to chaotic system forecasting and analysis tasks. For example, Pathak et al. (2017) propose a reservoir computing approach that fits the attractor of chaotic systems and predicts their Lyapunov exponents.

In this paper, we focus on such training strategies that require no architectural changes of the model and thus can be applied easily for different and existing sequence-to-sequence (seq2seq) models. All presented strategies will be evaluated across different benchmark datasets. Our main contributions are the following. First, assembling a set of training strategies for encoder-decoder RNNs that can be applied for existing seq2seq models without adapting their architecture. Second, presenting a collection of strategies' hyperparameter configurations that optimize the performance of the trained model. Third proposing a "flipped classroom" like strategy that outperforms all existing comparable approaches on several datasets sampled from different chaotic (and hyper-chaotic) systems. Fourth, proposing a method that yields substantially better prediction stability and therefore allows for forecasting longer horizons.

The course of the paper continues with Section 2 where we provide the background of sequence-to-sequence RNNs and the conventional ways to train them. We also give a short introduction to chaotic behavior here. In Section 3 we examine existing approaches dealing with or studying the *exposure bias* in the context of different applications. Section 4 describes how we designed our training strategies and how they are applied. Further information about the experimental setup and our results we present in Section 5. In Section 6 we discuss the results and the strengths and limitations of the different strategies. Finally, in Section 7 we conclude our findings.

## 2  Background

Within this paper, we study multi step forecasting of multivariate chaotic time series using RNNs. We are using an encoder-decoder architecture (Chung et al., 2014) as sequence-to-sequence model to forecast time series data. The rough structure of the encoder-decoder architecture is shown in Figure 1a. It consists of two

separate RNNs, an encoder and a decoder. The encoder is trained to build up a hidden state representing the recent history of the processed time series. It takes an input sequence $(x_1, x_2, \ldots, x_n)$ of $n$ input values where each value $x_j \in \mathbb{R}^d$ is a $d$ dimensional vector. For each processed step the encoder updates its hidden state to provide context information for the following steps. The last encoder hidden state (after processing $x_n$) is then used as initial hidden state of the decoder. Triggered by a sequence's preceding value, the decoder's task is predicting its next future value while taking into account the sequence's history via the accumulated hidden state. For a trained network in the inference phase, that means that the decoders preceding prediction is auto-regressively fed back as input into the decoder (aka auto-regressive prediction) (cp. Fig. 1a). All $m$ outputs $y_j \in \mathbb{R}^d$ together form the output sequence $(y_1, y_2, \ldots, y_m)$.

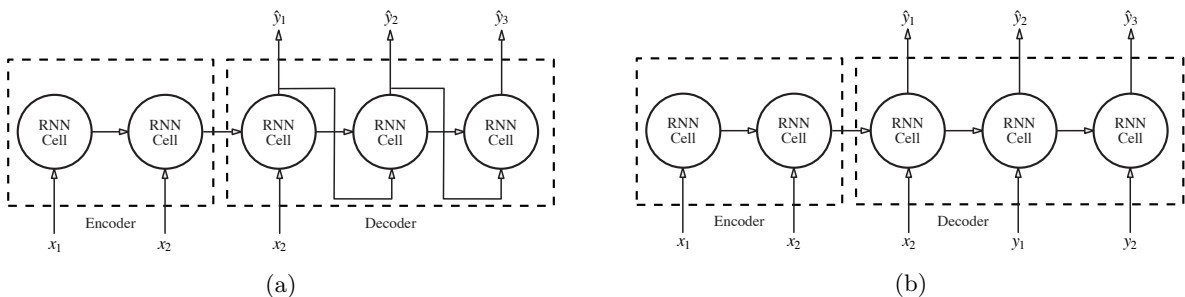

Figure 1: RNN encoder-decoder architecture in inference phase (a) and in training phase using teacher forcing (b)

## 2.1 Training Sequence Prediction Tasks

While training, the decoder's inputs may either be previous predicted outputs (free running) or known previous values stemming from the given dataset (teacher forcing). Training in free running mode might be an intuitive approach but slows down the training due to accumulated error throughout multiple prediction steps without the support of teacher forced inputs especially in the early training epochs (Nicolai & Silfverberg, 2020). In contrast, **teacher forcing** aims to avoid this problem by utilizing the corresponding previous ground truth value as decoder input rather than the previously predicted value. This way the model learns from the beginning of the training on to adapts its weights to perfect input values and converges Back noticeable faster (Miao et al., 2020) (cp. Fig. 1b). However, teacher forcing also bears a significant drawback since the model during training is never exposed to the noisy predicted values it will later face in the inference phase and it therefore does often not generalize very well. A model trained with teacher forcing solely learned to predict on basis of perfect values and is thus vulnerable for small perturbations on its input, this effect is called **exposure bias** (Ranzato et al., 2015).

## 2.2 Chaotic Systems

We especially focus on the forecasting of time series data generated from chaotic systems. Whether or not a dynamical system is chaotic, can be confirmed by considering its Lyapunov exponents $\lambda_k$ for $k \in [1, d]$. Given an initial perturbation $\varepsilon_0$ the exponential rate with which the perturbation will increase (or decrease) in the direction of dimension $i$ is the Lyapunov exponent $\lambda_k = \lim_{t \to \infty} \frac{1}{t} \ln(\frac{||\varepsilon_t||}{||\varepsilon_0||})$ (Dingwell, 2006). That is the Lyapunov exponents denote how sensitive the system is to the initial conditions (initial state). A deterministic dynamical system with at least one positive Lyapunov exponent while being aperiodic in its asymptotic limit is called **chaotic** (Dingwell, 2006). Analogously, dynamical systems with at least two positive, one negative, and one zero Lyapunov exponent are called **hyper-chaotic** systems. Dingwell points out that the Largest Lyapunov Exponent (LLE) can be used as a measure to compare the chaotic behavior of different systems.

## 3 Related Work

Schmidt (2019) defines *exposure bias* as describing "a lack of generalization with respect to an – usually implicit and potentially task and domain dependent – measure other than maximum-likelihood" meaning that when the exclusive objective is to maximize the likelihood between the output and the target sequence one can use teacher forcing during training (Goodfellow et al., 2016). However, Goodfellow et al. argue that the kind of input the model sees while testing will typically diverge from the training data and the trained model may lack the ability of correcting its own mistakes. Thus, in practice the teacher forcing can be a proper training strategy but may hinder the model to learn compensating its inaccuracies. He et al. study exposure bias for natural language generation tasks (He et al., 2019). They use sequence and word level quantification metrics to observe the influence of diverging prefix distributions on the distribution of the generated sequences. Two distributions are generated. One with and one without induced *exposure bias*. Those two distributions are then compared on basis of the corresponding corpus-bleu scores (Papineni et al., 2002). The study concludes that for language generation the effect of the *exposure bias* is less serious than widely believed.

As a result several studies propose approaches to overcome the *exposure bias* induced by teacher forcing. The earliest of these studies proposes **scheduled sampling** (Bengio et al., 2015). Scheduled sampling tries to take the advantages of training with teacher forcing while also acclimating the trained model to its own generated data distribution. It does that by using ground truth values as input for a subset of the training steps and predicted values for the remaining. $\epsilon_i$ denotes the teacher forcing probability at step $i$. Accordingly, the probability of using the predicted value is $1-\epsilon_i$. During training $\epsilon_i$ decreases from $\epsilon_s$ to $\epsilon_e$. This procedure makes it a **curriculum learning approach** as which scheduled sampling was proposed and works without major architectural adaptions. Originally proposed for image captioning tasks, scheduled sampling was also applied, e.g., for sound event detection (Drossos et al., 2019). Nicolai and Silfverberg consider and study the teacher forcing probability $\epsilon$ as a hyperparameter. Rather than using a decay function that determines the decrease of $\epsilon_i$ over the course of training epochs, they use a fix teacher forcing probability throughout the training (Nicolai & Silfverberg, 2020). They observed a moderate improvement compared to strict teacher forcing training.Scheduled sampling is not restricted to RNN-based sequence-to-sequence models though, it has also been studied for transformer architectures (Mihaylova & Martins, 2019). Mihaylova and Martins tested their modified transformer on two translation tasks but could only observe improved test results for one of them.

Apart from scheduled sampling (Bengio et al., 2015), a number of approaches have been proposed typically aiming to mitigate the *exposure bias* problem by adapting model architectures beyond an encoder-decoder design. **Professor forcing** (Lamb et al., 2016) is one of these more interfering approaches that aims to guide the teacher forced model in training by embedding it into a Generative Adversarial Network (GAN) framework (Goodfellow et al., 2014). This framework consists of two encoder-decoder RNNs that form the generator and the discriminator respectively. The generating RNNs have shared weights that are trained with the same target sequence using their respective inputs while at the same time they try to fool the discriminator by keeping their hidden states and outputs as similar as possible. The authors conclude that their method, compared to teacher forcing, provides better generalization for single and multi step prediction. In the field of Text-To-Speech (TTS), the concept of professor forcing has also been applied in the GAN based training algorithm proposed by Guo et al. (2019). They adapted professor forcing and found that replacing the teacher forcing generator with one that uses scheduled sampling improved the results of their TTS model in terms of intelligibility. As another approach for TTS, (Liu et al., 2020) proposed **teacher-student training** using a training scheme to keep the hidden states of the model in free running mode close to those of a model that was trained with teacher forcing. It applies a compound objective function to align the states of the teacher and the student model. The authors observe improved naturalness and robustness of the synthesized speech compared to their baseline. Dou et al. (Dou et al., 2019) proposed **attention forcing** as yet another training strategy for sequence-to-sequence models relying on an attention mechanism that forces a reference attention alignment while training the model without teacher forcing. They studied TTS tasks and observed a significant gain in quality of the generated speech. The authors conclude that attention forcing is especially robust in cases where the order of predicted output is irrelevant.

The discussed approaches for mitigating *exposure bias* were proposed in the context of NLP and mainly target speech or text generation. Additionally, these studies, except for scheduled sampling and subsequent approaches, mainly propose solutions that alter the model architecture to overcome *exposure bias* with potential other side effects for the actual task to train. Specifically for chaotic time series data, studies suggest to neglect teacher forcing completely and solely train the model in free running mode (Sangiorgio & Dercole, 2020), thereby, sacrificing the faster convergence of teacher forcing and potentially not reaching convergence at all. We argue that forecasting dynamical systems is a field that is not thoroughly dealt with regarding the mitigation of *exposure bias* and deserves more attention.

In the context of RNNs for forecasting and analyzing dynamical systems, the majority of existing work deals with exploding and vanishing gradients as well as capturing long-term dependencies while preserving the expressiveness of the network. Various studies rely on methods from dynamical systems theory applied to RNN or propose new network architectures.

Lusch et al. (2018) and Champion et al. (2019) use a modified autoencoder to learn appropriate eigenfunctions that the Koopman operator needs to linearize the nonlinear dynamics of the system. In another study, Vlachas et al. (2018) extend an LSTM model with a mean stochastic model to keep its state in the statistical steady state and prevent it from escaping the system's attractor. Schmidt et al. (2019) propose a more generalized version of a PLRNN (Koppe et al., 2019) by utilizing a subset of regularized memory units that hold information much longer and can thus keep track of long-term dependencies while the remaining parts of the architecture are designated to approximate the fast scale dynamics of the underlying dynamical system. The Antisymmetric Recurrent Neural Network (AntisymmetricRNN) introduced by Chang et al. (2019) represents an RNN designed to inherit the stability properties of the underlying ordinary differential equation (ODE) ensuring trainability of the network together with its capability of keeping track of long-term dependencies. A similar approach has been proposed as Coupled Oscillatory Recurrent Neural Networks (coRNNs) (Rusch & Mishra, 2020) that are based on a secondary order ODEs modeling a coupled network of controlled forced and damped nonlinear oscillators. The authors prove precise bounds of the RNN's state gradients and thus the ability of the coRNN being a possible solution for exploding or vanishing gradients. Erichson et al. (2020) propose the Lipschitz RNN having additional hidden-to-hidden matrices enabling the RNN to remain Lipschitz continuous. This stabilizes the network and alleviates the exploding and vanishing gradient problem. In Li et al. (2020; 2021), the authors propose the Fourier respectively the Markov neural operator that are built from multiple concatenated Fourier layers that directly work on the Fourier modes of the dynamical system. This way they retain major portion of the dynamics and forecast the future behavior of the system. Both, the incremental Recurrent Neural Network (IRNN) (Kag et al., 2019) and the time adaptive RNN (Kag & Saligrama, 2021) use additional recurrent iterations on each input to enable the model of coping different input time scales, where the later provides a time-varying function that adapts the model's behavior to the time scale of the provided input.

All of this shows the increasing interest in the application of machine learning (ML) models for forecasting and analyzing (chaotic) dynamical systems. To meet this trend, Gilpin (2021) recently published a fully featured collection of benchmark datasets being related to chaotic systems including their mathematical properties.

A more general guide of training RNNs for chaotic systems is given by Monfared et al. (2021). They discuss under which conditions the chaotic behavior of the input destabilizes the RNN and thus leads to exploding gradients during training. As a solution they propose STF, where every $\tau$-th time step a true input value is provided (teacher forced) as input instead of the previous prediction.

## 4 Teaching Strategies

Within this section, we systematically discuss existing teaching strategies for sequence-to-sequence prediction models and propose new strategies. All of these will then be evaluated in an experimental study with different chaotic time series reported in the following section.

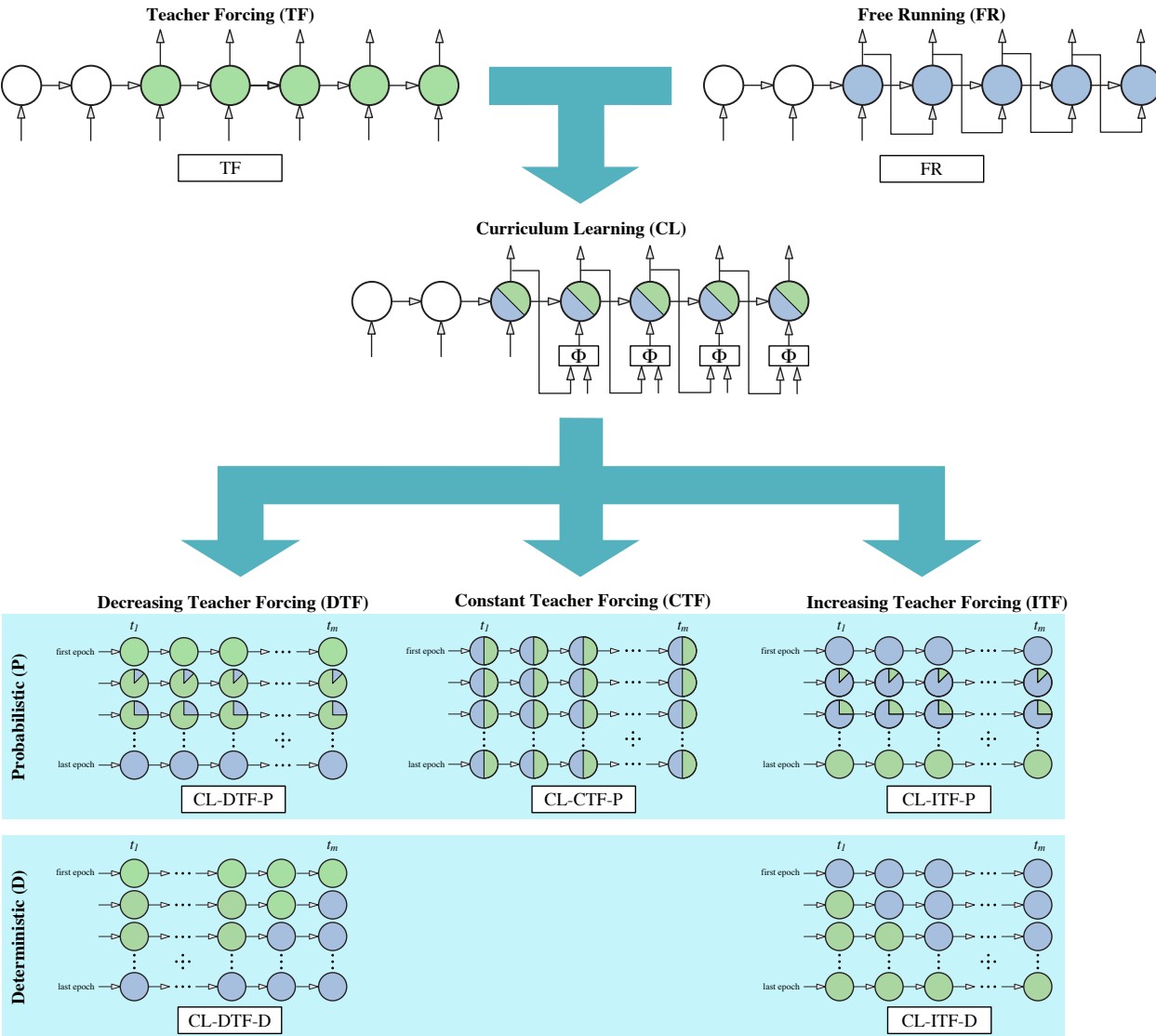

Figure 2: Overview of the proposed and evaluated training strategies where teacher forcing (TF) and free running (FR) refer to the two extreme cases that are combined to different Curriculum Learning (CL) strategies.

## 4.1 Free Running (FR) vs. Teacher Forcing (TF)

A widely used training strategy for RNN sequence-to-sequence models is to use teacher forcing (TF) throughout the whole training. Thereby, data is processed as shown in Fig. 2 (top left), i.e., the model is never exposed to its own predictions during training. A single forward step of the decoder during TF training is denoted as

$$\hat{y}^t = f(y^{t-1}, \theta, c^{t-1}), \tag{1}$$

where $y^t$ is the ground truth value for time step $t$, $\hat{y}^t$ is the predicted value for time step $t$, $\theta$ denotes the trainable parameters of the model, and $c^t$ is the decoder's hidden state at time step $t$.

The other extreme form of training is free running (FR), i.e., only the model's first output value is predicted on basis of ground truth input and all subsequent output values of the sequence are predicted on basis of previous predictions throughout the training (cp. Fig. 2 (top right)). A single forward step of the decoder

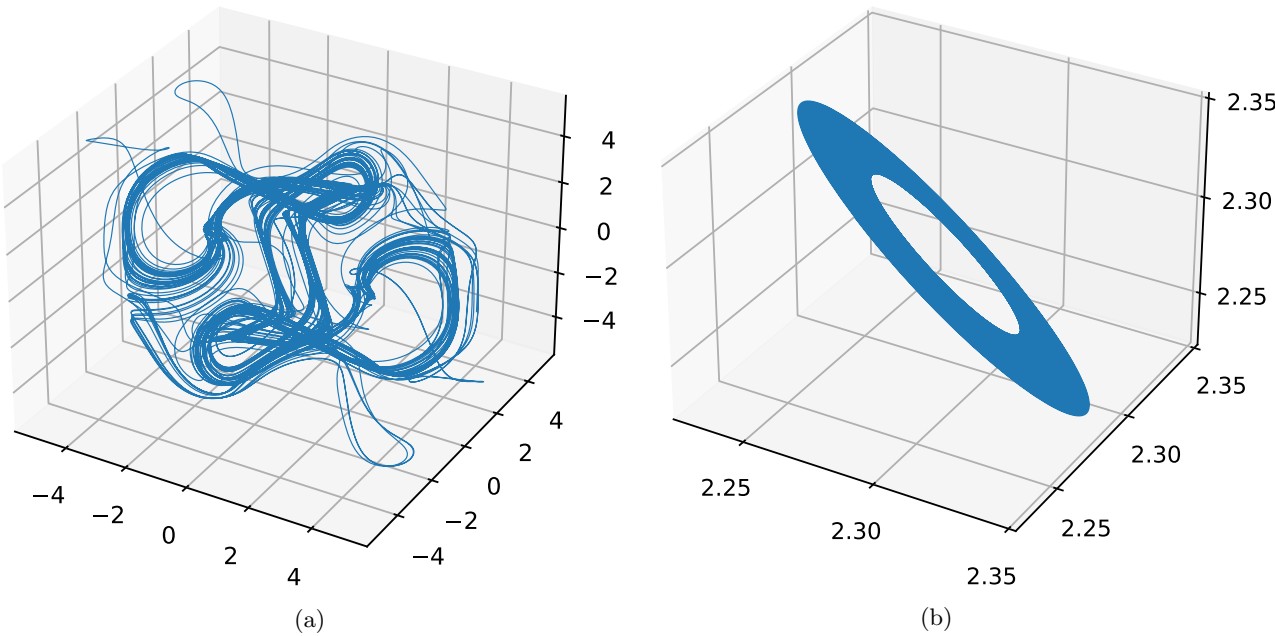

(a)                                                 (b)

Figure 3: 30 000 time steps sampled with a time delta of $dt = 0.1$ of Thomas' cyclically symmetric attractor in (a) a chaotic parametrization and (b) a periodic parametrization

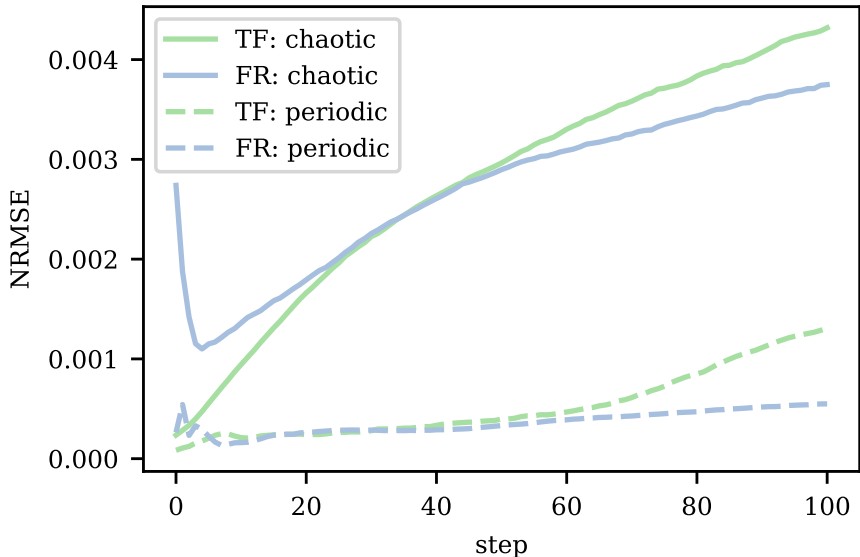

Figure 4: Test NRMSE over 100 predicted time steps of the chaotically and periodically parametrized Thomas attractor (cp. Fig. 3b), predicted by GRU models trained with teacher forcing (TF) and free running (FR).

during FR training is denoted as

$$\hat{y}^t = f(\hat{y}^{t-1}, \theta, c^{t-1}). \tag{2}$$

A claimed major benefit of TF training is faster model convergence and thus reduced training time (Miao et al., 2020), while a major benefit of FR training is avoided *exposure bias* arising from solely training with ground truth data yielding a model that performs less robust on unseen validation data (Ranzato et al., 2015). To illustrate these benefits and drawbacks, we utilize the Thomas attractor with two parametrization,

the first resulting in a periodic (cp. Fig. 3b) and the second resulting in a chaotic attractor (cp. Fig. 3a). By sampling from the attractors, we build two corresponding datasets of 10 000 samples each. For both datasets, we train a single layer encoder-decoder GRU following the free running (FR) and the teacher forcing (TF) strategy. Figure 4 shows test Normalized Root Mean Squared Error (NRMSE) of per trained model over 100 predicted time steps. All models have been initialized with 150 ground truth values to build up the hidden state before predicting these 100 time steps. We observe that the chaotic attractor is harder to predict for the trained models (cp. blue and green line in the figure) and that models trained with teacher forcing tend to predict with a smaller error at the first steps, which then grows relatively fast. In contrast, the prediction error of the FR trained models starts on a higher level but stays more stable over the prediction horizon. Arguing that chaotic time series forecasting represents an especially challenging task for sequence-to-sequence models, our work focuses on this type of time series data. The more precise forecasting capabilities of a TF-trained network at the early prediction steps vs. the overall more stable long-term prediction performance of a FR-trained network observed in the Thomas example (cp. Fig 3a, 3b), motivate the idea of combining both strategies into a curriculum learning approach.

Schmidt (2019) describes the *exposure bias* in natural language generation as a lack of generalization. Following this argumentation motivates an analysis of training with FR and TF strategies when applied to forecasting dynamical systems with different amounts of available training data. Figure 5 shows the NRMSE when forecasting the Thomas attractor using different dataset sizes and reveals that increasing the dataset size yields generally improved model performance for TF as well as FR, while their relative difference is maintained.

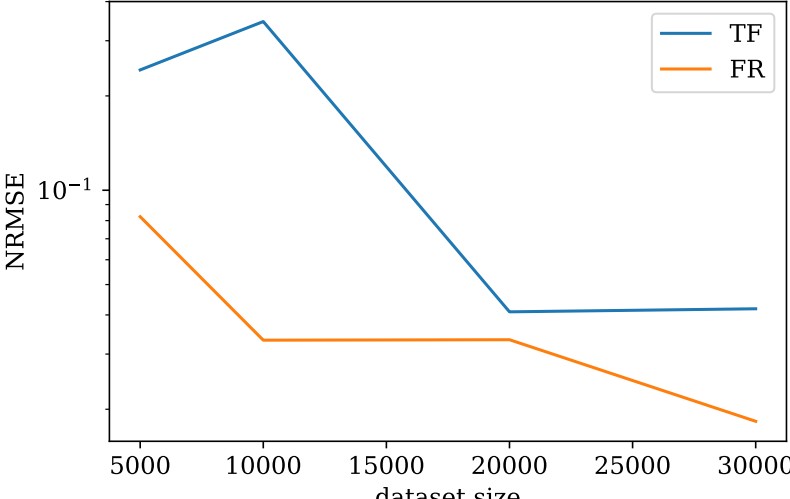

Figure 5: The NRMSE for different dataset sizes when using TF and FR during training for forecasting the Thomas attractor

## 4.2 Curriculum Learning (CL)

Within the context of our work, we denote the curriculum learning concept as combining teacher forcing and free running training, i.e., starting from the second decoder step the curriculum prescribes per decoder step whether to use the ground truth value or the predicted value of the previous time step as input. We formalize a single training step of a CL approach as follows:

$$\hat{y}^t = \begin{cases} f(y^{t-1}, \theta, c^{t-1}), & \text{if } \Phi = 1 \\ f(\hat{y}^{t-1}, \theta, c^{t-1}), & \text{otherwise} \end{cases} \tag{3}$$

where the teacher forcing decision $\Phi$ governs whether the decoder input is teacher forced or not. Figure 2 illustrates the data flow of a sequence-to-sequence model training with CL in between the conventional strategies. In our naming scheme CL-DTF-P resembles the scheduled sampling approach proposed by Bengio et al. (2015). Below, we discuss the different types of curricula on training and iteration scale resulting in different ways for determining $\Phi$.

## 4.3 Curriculum on Training Scale

The teacher forcing ratio $\epsilon_i$ per training iteration $i$ is determined by a curriculum function $C : \mathbb{N} \to [0, 1]$ denoted as

$$\epsilon_i = C(i). \tag{4}$$

We distinguish three fundamental types of curriculum on training scale. First, constant curricula where a constant amount of teacher forcing is maintained throughout the training denoted as

$$\epsilon_i = \epsilon. \tag{5}$$

Second, decreasing curricula where the training starts with a high amount of teacher forcing that continuously declines throughout the training. Third, increasing curricula where the training starts at a low amount of teacher forcing that continuously increases throughout the training. Both follow a transition function $C : \mathbb{N} \to [\epsilon_{start}, \epsilon_{end}]$ denoted as

$$\epsilon_i = C(i), \tag{6}$$

where $\epsilon_{start} \leq \epsilon_i \leq \epsilon_{i+1} \leq \epsilon_{end}$ for increasing curricula, $\epsilon_{start} \geq \epsilon_i \geq \epsilon_{i+1} \geq \epsilon_{end}$ for decreasing curricula and $\epsilon_{start} \neq \epsilon_{end}$ for both. The following equations exemplarily specify decreasing curricula (cp. Eqs. 7–9) following differing transition functions inspired by those used to study the scheduled sampling approach (Bengio et al., 2015)

$$
\begin{aligned}
C_{lin}(i) &= \max(\epsilon_{end}, \epsilon_{end} + (\epsilon_{start} - \epsilon_{end}) \cdot (1 - \frac{i}{Ł})), \\
&\quad \text{with } \epsilon_{end} < \frac{Ł - 1}{Ł}, \quad 1 < Ł, \quad i \in \mathbb{N}, \tag{7} \\
C_{invSig}(i) &= \epsilon_{end} + (\epsilon_{start} - \epsilon_{end}) \cdot \frac{k}{k + e^{\frac{i}{k}}}, \\
&\quad \text{with } \epsilon_{end} < \epsilon_{start}, \quad 1 \leq k, \quad i \in \mathbb{N}, \tag{8} \\
C_{exp}(i) &= \epsilon_{end} + (\epsilon_{start} - \epsilon_{end}) \cdot k^i, \\
&\quad \text{with } \epsilon_{end} < \epsilon_{start}, \quad 0 < k < 1, \quad i \in \mathbb{N}, \tag{9}
\end{aligned}
$$

where the curriculum length parameter $Ł$ determines the pace as number of iterations which the curriculum $C_{lin}$ needs to transition from $\epsilon_{start}$ to $\epsilon_{end}$. The curricula $C_{invSig}$ and $C_{exp}$ have no such parameter since the functions never completely reach $\epsilon_{end}$ in theory. In practice though, we adapt the curriculum specific parameter $k$ to stretch or compress these curricula along the iteration axis to achieve the same effect. Figure 6a exemplarily visualizes three decreasing and three increasing curricula following differing transition functions $C$ and being parametrized with $\epsilon_{start} = 1$ and $\epsilon_{end} = 0$ and $\epsilon_{start} = 0$ and $\epsilon_{end} = 1$ respectively. Furthermore, each is parametrized to have a curriculum length of $Ł = 1\,000$. Figure 6b shows examples of decreasing and increasing $C_{lin}$ with different $Ł$.

## 4.4 Curriculum on Iteration Scale

$\epsilon_i$ prescribes a ratio of TF vs. FR steps for a given training iteration $i$. Based on $\epsilon$ that solely prescribes the amount of teacher forcing for an iteration, we can now develop micro curricula for distributing the TF and FR steps eventually providing a teacher forcing decision $\Phi$ per training step. We propose two ways to distribute TF and FR steps within one training iteration: (1) probabilistic – where $\epsilon$ is interpreted as the probability of being a TF step, and (2) deterministic – where $\epsilon$ as a rate that determines the number of TF

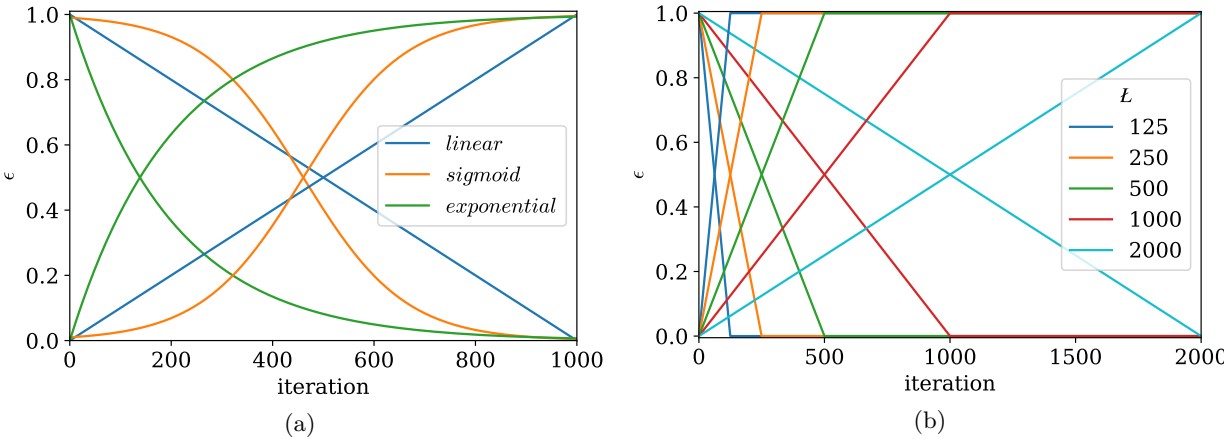

Figure 6: Examples of different decreasing curricula and their corresponding increasing versions (a) and multiple linear curricula with different pace Ł (b).

steps trained before moving to FR for the rest of the training sequence. For a probabilistic CL, we denote the teacher forcing decision $\Phi_\epsilon$, which is a discrete random variable that is drawn from a Bernoulli distribution:

$$\Phi_\epsilon \sim \text{Bernoulli}(\epsilon). \tag{10}$$

For a deterministic CL, $\Phi$ depends not only on $\epsilon$ but also on the current position $j$ within the predicted sequence of length $m$. Therefore, in this case we denote the teacher forcing decision $\Phi_{\epsilon,j}$ as:

$$\Phi_{\epsilon,j} = \begin{cases} 1, & \text{if } \epsilon \geq \frac{j}{m} \\ 0, & \text{otherwise.} \end{cases} \tag{11}$$

## 5 Evaluation

To compare the training strategies described in Section 4, we evaluate each with varying parametrization on six different chaotic time series' datasets. Our experiments aim to answer the following six research questions:

**RQ1 Baseline teaching strategies.** How well and consistent do the current baseline strategies FR and TF train a model for forecasting dynamical systems?

**RQ2 Curriculum learning strategies.** How do the different curriculum learning strategies perform in comparison to the baseline strategies?

**RQ3 Training length.** How is training length influenced by the different teaching strategies?

**RQ4 Prediction stability.** How stable is a model's prediction performance over longer prediction horizons when trained with the different strategies?

**RQ5 Curriculum parametrization.** How much does the curriculum's parametrization influence model performance?

**RQ6 Iteration scale curriculum.** How do iteration scale curricula differ in resulting model performance?

### 5.1 Evaluated Curricula

In total, we define eight strategies that we evaluate in this study (cp. Fig. 2). For comparison, we train the two baseline methods teacher forcing (TF) and free running (FR) that "teach" throughout the entire training or do not "teach" at all respectively. All other methods prescribe a teaching curriculum CL throughout the training and we distinguish these strategies along two dimensions: (1) the overall increasing (ITF), constant (CTF), or decreasing (DTF) trend in teacher forcing throughout the training curriculum and (2) the probabilistic (P) or deterministic (D) teacher forcing distribution within training steps.

Table I: Curriculum strategy parameters used during the *baseline*, *exploratory*, and the *essential* experiments

|  | Strategy | parameter | Curriculum evaluated values |
|---|---|---|---|
| *baseline* | FR | $C$ | – |
|  |  | $\epsilon$ | 0.0 |
|  |  | Ł | – |
|  | TF | $C$ | – |
|  |  | $\epsilon$ | 1.0 |
|  |  | Ł | – |
| *exploratory* | CL-CTF-P | $C$ | – |
|  |  | $\epsilon$ | $\{0.25, 0.5, 0.75\}$ |
|  |  | Ł | – |
|  | CL-DTF-P, CL-DTF-D | $C$ | $\{\text{linear, inverse sigmoid, exponential}\}$ |
|  |  | $\epsilon_{start} \rightarrow \epsilon_{end}$ | $\{0.25, 0.5, 0.75, 1.0\} \rightarrow 0.0$ |
|  |  | Ł | 1000 |
|  | CL-ITF-P, CL-ITF-D | $C$ | $\{\text{linear, inverse sigmoid, exponential}\}$ |
|  |  | $\epsilon_{start} \rightarrow \epsilon_{end}$ | $\{0.0, 0.25, 0.5, 0.75\} \rightarrow 1.0$ |
|  |  | Ł | 1000 |
| *essential* | CL-DTF-P, CL-DTF-D, CL-ITF-P, CL-ITF-D | $C$ | linear |
|  |  | $\epsilon_{start} \rightarrow \epsilon_{end}$ | $\{0.0 \rightarrow 1.0, 1.0 \rightarrow 0.0\}$ |
|  |  | Ł | $\{62, 125, 250, 500, 1\,000, 2\,000,$ $4\,000, 8\,000, 16\,000, 32\,000\}$ |

### 5.2 Parametrization of Training Curricula

Table I shows all training strategy-specific parameters and their values for the evaluated strategies. We subdivide our experiments into three sets: *baseline*, *exploratory*, and *essential* experiments. The baseline strategies FR and TF do not have any additional parameters. The CL-CTF-P strategy has the $\epsilon$ parameter configuring the strategy's teacher forcing ratio. The increasing and decreasing strategies CL-DTF-x and CL-ITF-x are configured by $\epsilon_{start}$ and $\epsilon_{end}$ referring to the initial and eventual amount of teacher forcing and the function $C$ transitioning between both. Additionally, Ł determines the number of training epochs in between $\epsilon_{start}$ and $\epsilon_{end}$. For the *exploratory* experiments, we utilize a fix Ł $= 1\,000$, while for the *essential* experiments, we evaluate all strategies solely using a linear transition $C_{linear}$ in the curriculum (cp. Eq. 7) with either $\epsilon_{start} = 0$ and $\epsilon_{end} = 1$ (increasing) or $\epsilon_{start} = 1$ and $\epsilon_{end} = 0$ (decreasing).

### 5.3 Performance Metrics

We use the NRMSE and the $R^2$ metrics as well as two derived of those to evaluate model performance. NRMSE is a normalized version of the Root Mean Squared Error (RMSE) where smaller values indicate

Table II: Details of the chaotic systems that were approximated to generate the data used for our experiments

| System | ODE/DDE | Parameters | $d$ | LLE |
|---|---|---|---|---|
| Mackey-Glass | $\frac{dx}{dt} = \beta \frac{x_\tau}{1+x_\tau^n} - \gamma x$ with $\gamma, \beta, n > 0$ | $\tau = 17$, $n = 10$, $\gamma = 0.1$ $\beta = 0.2$, $dt = 1.0$ | 1 | 0.006 |
| Thomas | $\frac{dx}{dt} = sin(y) - bx$ $\frac{dy}{dt} = sin(z) - by$ $\frac{dz}{dt} = sin(x) - bz$ | $b = 0.1$, $dt = 0.1$ | 3 | 0.055 |
| Rössler | $\frac{dx}{dt} = -(y + z)$ $\frac{dy}{dt} = x + ay$ $\frac{dz}{dt} = b + z(x - c)$ | $a = 0.2$, $b = 0.2$ $c = 5.7$, $dt = 0.12$ | 3 | 0.069 |
| Hyper Rössler | $\frac{dx}{dt} = -y - z$ $\frac{dy}{dt} = x + ay + w$ $\frac{dz}{dt} = b + xz$ $\frac{dw}{dt} = -cz + dw$ | $a = 0.25$, $b = 3$ $c = 0.5$, $d = 0.05$ $dt = 0.1$ | 4 | 0.14 |
| Lorenz | $\frac{dx}{dt} = -\sigma x + \sigma y$ $\frac{dy}{dt} = -xz + \rho x - y$ $\frac{dz}{dt} = xy - \beta z$ | $\sigma = 10$, $\beta = \frac{8}{3}$ $\rho = 28$, $dt = 0.01$ | 3 | 0.905 |
| Lorenz'96 | $\frac{dx_k}{dt} = -x_{k-2}x_{k-1} + x_{k-1}x_{k+1} - x_k + F$ for $k = 1 \ldots d$ and $x_{-1} = x_d$ | $F = 8$, $dt = 0.05$ | 40 | 1.67 |

better prediction performance. For a single value of a sequence, NRMSE is calculated as:

$$\text{NRMSE}(y, \hat{y}) = \frac{\sqrt{\frac{1}{d} \cdot \sum_{j=1}^{d}(y_j - \hat{y}_j)^2}}{\sigma}, \tag{12}$$

where $y$ is a ground truth vector, $\hat{y}$ is the corresponding prediction, $\sigma$ is the standard deviation across the whole dataset, and $d$ is the size of the vectors $y$ and $\hat{y}$. For model evaluation, we calculate the mean NRMSE over all $m$ forecasted steps of a sequence. Additionally, we compute and report the NRMSE only for the last $\lceil \frac{m}{10} \rceil$ forecasted steps of a sequence to specifically evaluate model performance at long prediction horizons.

The $R^2$ score lies in the range $(-\infty, 1]$ with higher values referring to better prediction performance. A score of 0 means that the prediction is as good as predicting the ground truth sequence's mean vector $\bar{y}$. The $R^2$ score is computed as:

$$R^2 = 1 - \frac{\sum_{j=1}^{d}(y_j - \hat{y}_j)^2}{\sum_{j=1}^{d}(y_j - \bar{y}_j)^2}. \tag{13}$$

We use the $R^2$ score to compute another metric $\text{LT}R^2 > 0.9$ measuring the number of Lyapunov Time (LT)s that a model can predict without the $R^2$ score dropping below a certain a threshold of 0.9. Sangiorgio and Dercole (Sangiorgio & Dercole, 2020) proposed this metric while applying a less strict threshold of 0.7.

## 5.4 Evaluated Datasets

We focus on forecasting chaotic time series data and sample datasets by approximating six commonly studied chaotic systems (cp. Tab. II), i.e., Mackey-Glass (Mackey & Glass, 1977), Rössler (Rössler, 1976), Thomas' cyclically symmetric attractor (Thomas, 1999), Hyper Rössler (Rossler, 1979), Lorenz (Lorenz, 1963) and Lorenz'96 (Lorenz, 1996). Table II shows per system, the differential equations and how we parametrized them. These systems, differ among other in the number of dimensions $d$ and degree of chaos as indicated by the largest lyapunov exponent in the LLE column of Tab. II. The LLEs are approximated values that were published independently in the past (Brown et al., 1991; Sprott & Chlouverakis, 2007; Sano & Sawada, 1985; Sandri, 1996; Hartl, 2003; Brajard et al., 2020). We generate datasets by choosing an initial state vector

Table III: Results of the *exploratory* tests with the best hyperparameter configuration per strategy and system. The arrow besides each metric's column title indicates whether smaller (↓) or larger (↑) values are favored. The best result values per dataset are printed in bold and the best baseline NRMSEs are underlined. Together with each dataset we put the corresponding LLE in parenthesis.

| | Strategy | Best performing curriculum $C$ | $\epsilon$ | Trained epochs | NRMSE over 1LT absolut ↓ | rel. impr. ↑ | @BL epoch ↓ | last 10% ↓ |
|---|---|---|---|---|---|---|---|---|
| **Thomas (0.055)** | FR | constant | 0.00 | 427 | 0.03416 | – | – | 0.047222 |
| | TF | constant | 1.00 | 163 | 0.34545 | – | – | 0.607954 |
| | CL-CTF-P | constant | 0.25 | 450 | 0.05535 | −62.03% | 0.05675 | 0.082443 |
| | CL-DTF-P | inverse sigmoid | 0.75 ↘ 0.00 | 598 | 0.01858 | 45.61% | 0.02120 | 0.034325 |
| | CL-DTF-D | exponential | 0.25 ↘ 0.00 | 557 | 0.03229 | 5.47% | 0.03792 | 0.039749 |
| | CL-ITF-P | exponential | 0.00 ↗ 1.00 | 620 | 0.01403 | 58.93% | **0.02026** | 0.026014 |
| | CL-ITF-D | exponential | 0.25 ↗ 1.00 | 944 | **0.01126** | **67.04%** | 0.02179 | **0.018571** |
| **Rössler (0.069)** | FR | constant | 0.00 | 3 863 | 0.00098 | – | – | 0.000930 |
| | TF | constant | 1.00 | 500 | 0.00743 | – | – | 0.016119 |
| | CL-CTF-P | constant | 0.25 | 2 081 | 0.00084 | 14.29% | 0.00084 | 0.001333 |
| | CL-DTF-P | linear | 1.00 ↘ 0.00 | 2 751 | 0.00083 | 15.31% | 0.00083 | 0.000931 |
| | CL-DTF-D | inverse sigmoid | 0.25 ↘ 0.00 | 4 113 | 0.00064 | 34.69% | 0.00066 | 0.000578 |
| | CL-ITF-P | inverse sigmoid | 0.00 ↗ 1.00 | 7 194 | 0.00025 | 74.49% | 0.00034 | **0.000358** |
| | CL-ITF-D | linear | 0.75 ↗ 1.00 | 5 132 | **0.00024** | **75.51%** | **0.00031** | 0.000390 |
| **Lorenz (0.905)** | FR | constant | 0.00 | 918 | 0.01209 | – | – | 0.013166 |
| | TF | constant | 1.00 | 467 | 0.00152 | – | – | 0.002244 |
| | CL-CTF-P | constant | 0.75 | 297 | 0.00167 | −9.87% | 0.00167 | 0.002599 |
| | CL-DTF-P | inverse sigmoid | 0.75 ↘ 0.00 | 522 | 0.00168 | −10.53% | 0.00162 | 0.002425 |
| | CL-DTF-D | inverse sigmoid | 1.00 ↘ 0.00 | 204 | 0.00187 | −23.03% | 0.00187 | 0.002823 |
| | CL-ITF-P | linear | 0.00 ↗ 1.00 | 750 | 0.00149 | 1.97% | 0.00217 | 0.002235 |
| | CL-ITF-D | inverse sigmoid | 0.75 ↗ 1.00 | 803 | **0.00124** | **18.42%** | **0.00132** | **0.002084** |
| **Lorenz'96 (1.67)** | FR | constant | 0.00 | 8 125 | 0.07273 | – | – | 0.126511 |
| | TF | constant | 1.00 | 4 175 | 0.03805 | – | – | 0.075583 |
| | CL-CTF-P | constant | 0.50 | 2 615 | 0.07995 | −110.12% | 0.07995 | 0.140700 |
| | CL-DTF-P | linear | 0.75 ↘ 0.00 | 939 | 0.04654 | −22.31% | 0.04654 | 0.087228 |
| | CL-DTF-D | linear | 0.75 ↘ 0.00 | 1 875 | 0.04381 | −15.14% | 0.04381 | 0.081025 |
| | CL-ITF-P | inverse sigmoid | 0.25 ↗ 1.00 | 4 787 | **0.01854** | **51.27%** | **0.02016** | **0.036651** |
| | CL-ITF-D | inverse sigmoid | 0.00 ↗ 1.00 | 3 263 | 0.02093 | 44.99% | 0.02196 | 0.040356 |

of size $d$ and approximate 10 000 samples using the respective differential equations. We use SciPy package's implementation of the Livermore solver for ordinary differential equations (LSODE) (Radhakrishnan & Hindmarsh, 1993) except for Mackey-Glass which we approximate through the Python module JiTCDDE implementing the delayed differential equation (DDE) integration method proposed by (Shampine & Thompson, 2001). Thereby, $dt$ defines the time difference between two sampled states per dataset and is shown in Table II. Where available, we chose $dt$ similar to previous studies aiming for comparability of results. We split each dataset into 80% training samples and 10% validation and testing samples respectively. All data is normalized following a $z$-transform.

## 5.5 Training Procedure

All evaluated models follow an encoder-decoder GRU architecture with an additional fully connected layer after the decoder (cp. Fig. 7). We performed a full grid search for the hyper-parameters learning rate, batch size, learning rate reduction factor, loss plateau, input length $n$ and hidden state size to determine suitable configurations for the experiments. Based on this optimization, we used the Adam (Kingma et al., 2015) optimizer with a batch size of 128 and apply Reduce Learning Rate on Plateau (RLROP) with an initial learning rate of $1e^{-3}$ and a reduction factor of 0.6, i.e., 40% learning rate reduction, given a loss plateau of 10 epochs for all datasets except Lorenz'96 where we use a reduction factor of 0.9 and a 20 epoch plateau respectively. Furthermore, we found an input length of $n = 150$ steps and a *hidden state size* of 256 to be

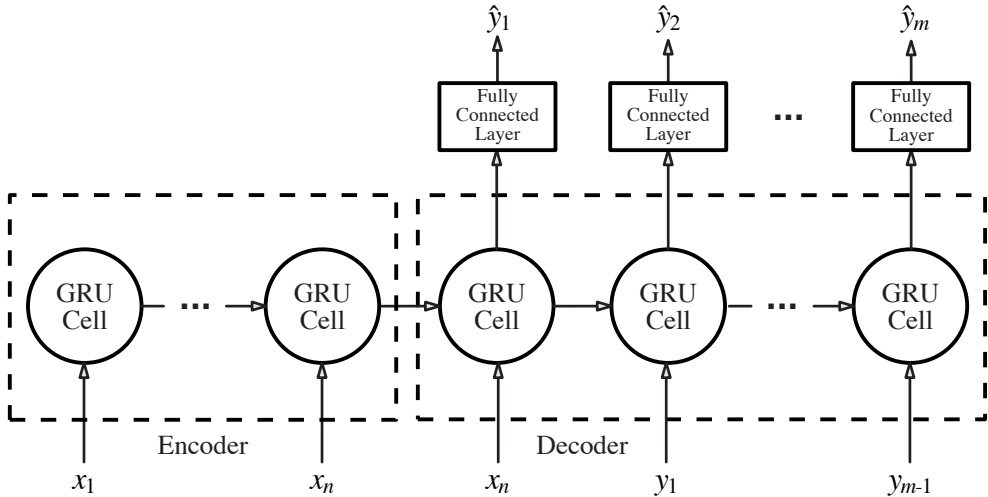

Figure 7: Structure of a simple encoder-decoder GRU used for training with teacher forcing

most suitable. We use early stopping with a *patience* of 100 epochs and a *minimum improvement threshold* of 1% to ensure the convergence of the model while preventing from overfitting. We train all models with a dataset-specific prediction length $m$ defined as:

$$m = \left\lceil \frac{\text{LT}}{dt} \right\rceil = \left\lceil \frac{1}{dt \cdot \text{LLE}} \right\rceil. \tag{14}$$

The reason being that we aim to train for the same forecasting horizon that we mainly evaluate a trained model with. We adapt this horizon to the dataset's LT thereby aiming for performance measures that are comparable across datasets.

We provide plots of the training and validation loss curves of the final parametrization per strategy and dataset in Appendix A.1. Based on these curves, we observe for ITF in contrast to DTF strategies that the training loss tends to move away from the validation loss faster. This is explainable by the fact that with increasing training time ITF strategies deliver an increasing amount of TF inputs counteracting the accumulation of error along the forecasted sequence and therefore further reducing training loss. For DTF strategies we observe an opposing behavior. Regarding training iterations, we observe that ITF strategies typically train for a larger number of epochs. Since the termination of the training is determined by the early stopping criterion this shows that ITF facilitates a longer and typically more successful training process compared to DTF and baseline strategies.

### 5.6 Results

Table III shows results for the *exploratory* experiments. Per evaluated strategy and dataset, the table reports resulting model performance in terms of NRMSE. We only report that curriculum configuration in terms of transition function $C$ and $\epsilon$ schedule that yields the highest NRMSE per strategy and dataset. Each model was used to predict a dataset-spetefic horizon of 1 LT. The best result per dataset and performance metric is highlighted in bold. First, we study the baseline strategies FR and TF and observe that for two datasets, i.e., Thomas and Rössler, the FR strategy outperforms the TF baseline, while TF outperforms FR for the other two. We select the one that performs best per dataset to measure relative improvement or deterioration gained by training with the respective curriculum learning strategy (cp. column "NMRSE rel. impr."). We observe that across all datasets and performance metrics the CL-ITF-P and CL-ITF-D strategies yield the best and second best performing model with a relative improvement of $1.97 - 80.61\%$ over the best performing baseline strategy. The other curriculum learning strategies perform less consistent across the datasets. The CL-DTF-x strategies yield an improved NRMSE for half of the datasets while the constant

Table IV: Best curriculum length per strategy and system for all six datasets. The arrow besides a metric's column title indicates whether smaller (↓) or larger (↑) values are favored. The best result values per dataset are printed in bold and the best baseline NRMSEs are underlined. Together with each dataset we put the corresponding LLE in parenthesis.

| | Strategy | Best performing curriculum $\epsilon$ | Ł | Trained epochs | NRMSE over 1LT absolut↓ | rel. impr.↑ | @BL epoch↓ | last 10%↓ | #LT with $R^2 > 0.9$↑ |
|---|---|---|---|---|---|---|---|---|---|
| Mackey Glass (0.006) | FR | 0.00 | – | 4713 | 0.00391 | – | 0.00391 | 0.004101 | 4.50 |
| | TF | 1.00 | – | 44 | 0.09535 | – | 0.09535 | 0.171945 | 1.64 |
| | CL-CTF-P | 0.25 | – | 2918 | 0.00632 | −61.64% | 0.00632 | 0.006544 | 4.51 |
| | CL-DTF-P | 1.00↘0.00 | 2000 | 3733 | 0.00215 | 45.01% | 0.00215 | 0.003010 | 4.95 |
| | CL-DTF-D | 1.00↘0.00 | 1000 | 431 | 0.00585 | −49.62% | 0.00585 | 0.011022 | 3.91 |
| | CL-ITF-P | 0.00↗1.00 | 500 | 1566 | **0.00104** | **73.40%** | **0.00104** | **0.001793** | **5.18** |
| | CL-ITF-D | 0.00↗1.00 | 500 | 1808 | 0.00211 | 46.03% | 0.00211 | 0.003032 | 4.99 |
| Thomas (0.055) | FR | 0.00 | – | 427 | 0.03416 | – | 0.03416 | 0.047222 | 2.04 |
| | TF | 1.00 | – | 163 | 0.34545 | – | 0.34545 | 0.607954 | 1.73 |
| | CL-CTF-P | 0.25 | – | 450 | 0.05535 | −62.03% | 0.05675 | 0.082443 | 1.53 |
| | CL-DTF-P | 1.00↘0.00 | 1000 | 356 | 0.05084 | −48.83% | 0.05084 | 0.105585 | 2.13 |
| | CL-DTF-D | 1.00↘0.00 | 1000 | 326 | 0.10712 | −213.58% | 0.10712 | 0.206923 | 1.53 |
| | CL-ITF-P | 0.00↗1.00 | 500 | 677 | **0.00930** | **72.78%** | **0.01645** | **0.016729** | **3.99** |
| | CL-ITF-D | 0.00↗1.00 | 500 | 649 | 0.01819 | 46.75% | 0.03934 | 0.030589 | 2.05 |
| Rössler (0.069) | FR | 0.00 | – | 3863 | 0.00098 | – | 0.00098 | 0.000930 | 9.46 |
| | TF | 1.00 | – | 500 | 0.00743 | – | 0.00743 | 0.016119 | 4.75 |
| | CL-CTF-P | 0.25 | – | 2081 | 0.00084 | 14.29% | 0.00084 | 0.001333 | 7.51 |
| | CL-DTF-P | 1.00↘0.00 | 1000 | 2751 | 0.00083 | 15.31% | 0.00083 | 0.000931 | 8.46 |
| | CL-DTF-D | 1.00↘0.00 | 125 | 4879 | 0.00100 | −2.04% | 0.00116 | 0.000947 | 9.28 |
| | CL-ITF-P | 0.00↗1.00 | 500 | 4523 | **0.00019** | **80.61%** | **0.00022** | **0.000303** | **10.23** |
| | CL-ITF-D | 0.00↗1.00 | 4000 | 7267 | 0.00027 | 72.24% | 0.00051 | 0.000368 | 9.41 |
| Hyper Rössler (0.14) | FR | 1.00 | – | 6461 | 0.00599 | – | 0.00599 | 0.007011 | 6.57 |
| | TF | 0.00 | – | 2788 | 0.00435 | – | 0.00762 | 0.011194 | 5.24 |
| | CL-CTF-P | 0.25 | – | 2909 | 0.01450 | −233.33% | 0.01450 | 0.015944 | 5.21 |
| | CL-DTF-P | 1.00↘0.00 | 2000 | 3773 | 0.00560 | 28.74% | 0.00560 | 0.007052 | 6.32 |
| | CL-DTF-D | 1.00↘0.00 | 16000 | 1793 | 0.00490 | 12.64% | 0.00490 | 0.007471 | 6.30 |
| | CL-ITF-P | 0.00↗1.00 | 125 | 2802 | 0.00366 | 15.86% | 0.00366 | 0.005802 | 6.50 |
| | CL-ITF-D | 0.00↗1.00 | 250 | 3317 | **0.00326** | **25.06%** | **0.00326** | **0.004639** | **6.72** |
| Lorenz (0.905) | FR | 0.00 | – | 918 | 0.01209 | – | 0.01319 | 0.013166 | 3.31 |
| | TF | 1.00 | – | 467 | 0.00152 | – | 0.00152 | 0.002244 | 6.72 |
| | CL-CTF-P | 0.75 | – | 297 | 0.00167 | −9.87% | 0.00167 | 0.002599 | 6.43 |
| | CL-DTF-P | 1.00↘0.00 | 4000 | 450 | 0.00124 | 18.42% | **0.00124** | 0.001925 | 6.64 |
| | CL-DTF-D | 1.00↘0.00 | 16000 | 587 | 0.00111 | 26.97% | 0.00127 | 0.001650 | 6.53 |
| | CL-ITF-P | 0.00↗1.00 | 250 | 1137 | **0.00060** | **60.53%** | **0.00124** | **0.000883** | **7.19** |
| | CL-ITF-D | 0.00↗1.00 | 250 | 578 | 0.00135 | 11.18% | 0.00189 | 0.001725 | 4.33 |
| Lorenz'96 (1.67) | FR | 0.00 | – | 8125 | 0.07273 | – | 0.08362 | 0.126511 | 2.34 |
| | TF | 1.00 | – | 4175 | 0.03805 | – | 0.03805 | 0.075583 | 3.01 |
| | CL-CTF-P | 0.50 | – | 2615 | 0.07995 | −110.12% | 0.07995 | 0.140700 | 2.25 |
| | CL-DTF-P | 1.00↘0.00 | 1000 | 983 | 0.05278 | −38.71% | 0.05278 | 0.098130 | 2.67 |
| | CL-DTF-D | 1.00↘0.00 | 1000 | 4083 | 0.07119 | −87.10% | 0.07119 | 0.126636 | 2.34 |
| | CL-ITF-P | 0.00↗1.00 | 250 | 3886 | 0.01680 | 55.85% | 0.01680 | 0.032439 | 4.01 |
| | CL-ITF-D | 0.00↗1.00 | 250 | 3379 | **0.01628** | **57.21%** | **0.01628** | **0.031464** | **4.18** |

CL-CTF-P only yields an improvement for the Thomas attractor. We separately report the NRMSE of the last 10% predicted values of 1 LT test horizon per dataset to assess how robust a prediction is over time (cp. column "NRMSE last 10%"). We observe that the CL-ITF-P and CL-ITF-D strategies also reach the best performance in terms of this metric meaning that they yield the most robust models. We further observe a diverse set of curriculum configurations yielding the best performing model per strategy and dataset. That means that all available transition functions, i.e., linear, inverse sigmoid, and exponential, have been discovered as best choice for at least one of the trained models. Further, we observe all evaluated $\epsilon$ as best choice for the CL-CTF-P strategy and one of the datasets respectively. Similarly, the best performing initial $\epsilon$ for the increasing and decreasing transitions per dataset spans all evaluated values except for 0.5. The table also reports the number of training iterations till reaching the early stopping criterion (cp. column "training

epochs"). We observe that the two baseline strategies utilize strongly differing numbers of iterations across all datasets. For the Thomas and the Rössler attractor, the teacher forcing strategy TF does not allow for proper model convergence being characterized by a low number of iterations and a high NRMSE compared to the other strategies. Among the curriculum teaching strategies across all datasets, the strategies with increasing teacher forcing ratio CL-ITF-x utilize the most training iterations. These CL-ITF-x strategies also utilize more training iterations than the better performing baseline strategy across all datasets. To better understand whether the longer training is the sole reason for the higher performance of the CL-ITF-x trained models, we additionally report the performance in terms of NRMSE of all curriculum trained models after the same number of training iterations as the better performing baseline model, i.e., after 427 epochs for Thomas, after 3 863 epochs for Rössler, after 467 epochs for Lorenz, and after 4 175 epochs for Lorenz'96 (cp. column "Performance @BL epochs"). We observe across all datasets that the best performing teaching strategy still remains CL-ITF-P or CL-ITF-D. In conclusion, the *exploratory* experiments demonstrated that a well-parametrized CL-ITF-x strategy yields a 18.42 – 75.51% performance increase across the evaluated datasets.

However, this improvement comes at the cost of an intensive parameter optimization of the respective curriculum. Therefore, we run a second series of *essential* experiments in which we simplify the parametrization of the curriculum by utilizing a linear transition from either $0.0 \rightarrow 1.0$ (CL-ITF-x) or $1.0 \rightarrow 0.0$ (CL-DTF-x) that is solely parametrized by the length of this transition in terms of training epochs Ł. Table IV reports results in terms of the previously introduced performance metrics again measured over a prediction horizon of 1 LT and across the same teaching strategies for six datasets including those studied for the *exploratory* experiments. Since the changes of the *essential* over the *exploratory* experiments solely effect teaching strategies with a training iteration-dependent curriculum, they have no effect on the baseline strategies FR and TF as well as the constant curriculum CL-CTF-P, which we still report in Table IV for direct comparison. Overall, we observe that CL-ITF-P outperforms all other strategies for four out of six datasets while it performs second best for the remaining two datasets where the deterministic version CL-ITF-D performs best. These strategies yield relative improvements ranging from 25.06 – 80.61% and are, thus, even higher than those observed for the *exploratory* experiments. Beyond that we observe that for all datasets treated in both experimental sets, the training curricula used in the *essential* experiments yield better performing models. For three out of four datasets the training even requires substantially less training iterations than in the explorative experiments. Additionally, we report in column "#LT with $R^2 > 0.9$" the prediction horizon in terms of LT that a trained model can predict while constantly maintaining an $R^2 > 0.9$. We observe that the ranking between the different strategies mostly remains the same as those observed when predicting 1 LT. That means that also the best performing strategy consistently remains the same for the longer horizon. Figure 8 more concretely depicts how the $R^2$ score develops across the prediction horizon for the different teaching strategies and datasets.

## 5.7 Additional Experiments

Judging from the essential experiments CL-ITF-x are our winning strategies on the chaotic systems we tested. However, as mentioned in Section 3 there are many other approaches targeting (chaotic) dynamical system forecasting with adapted RNNs architectures that take theoretical insights of dynamical systems into account. STF (Monfared et al., 2021) does not require any architectural modifications but instead provides an adapted training strategy. It determines a time interval $\tau = \frac{ln2}{LLE}$ that denotes how many FR steps are processed before the next TF value is used within one sequence. It is the strategy that we found most comparable to the CL approaches we study. Therefore, we executed another set of experiments where we used STF during the training of our encoder-decoder GRU for all chaotic systems in Table II. Since our data is sampled with different $dt$ we have to redefine the time interval as $\tau = \frac{ln2}{LLE \cdot dt}$.

The results (cp. Tab. V) show that STF provides improved performance compared to the best baseline for three of six datasets ranging from 26.21 – 46.75% relative improvement. For this it does require no additional hyperparameters if the systems LLE is known. It also beats the best performing CL strategy on the Hyper-Rössler dataset by a margin of 1.15%. For the rest of the datasets, the results stay behind those of the CL-ITF-x strategies showing a worse, i.e., increased, NRMSE by 7.00 – 236.18%. We assume that where STF systematically induces TF to catch chaos preventing exploding gradients before they appear

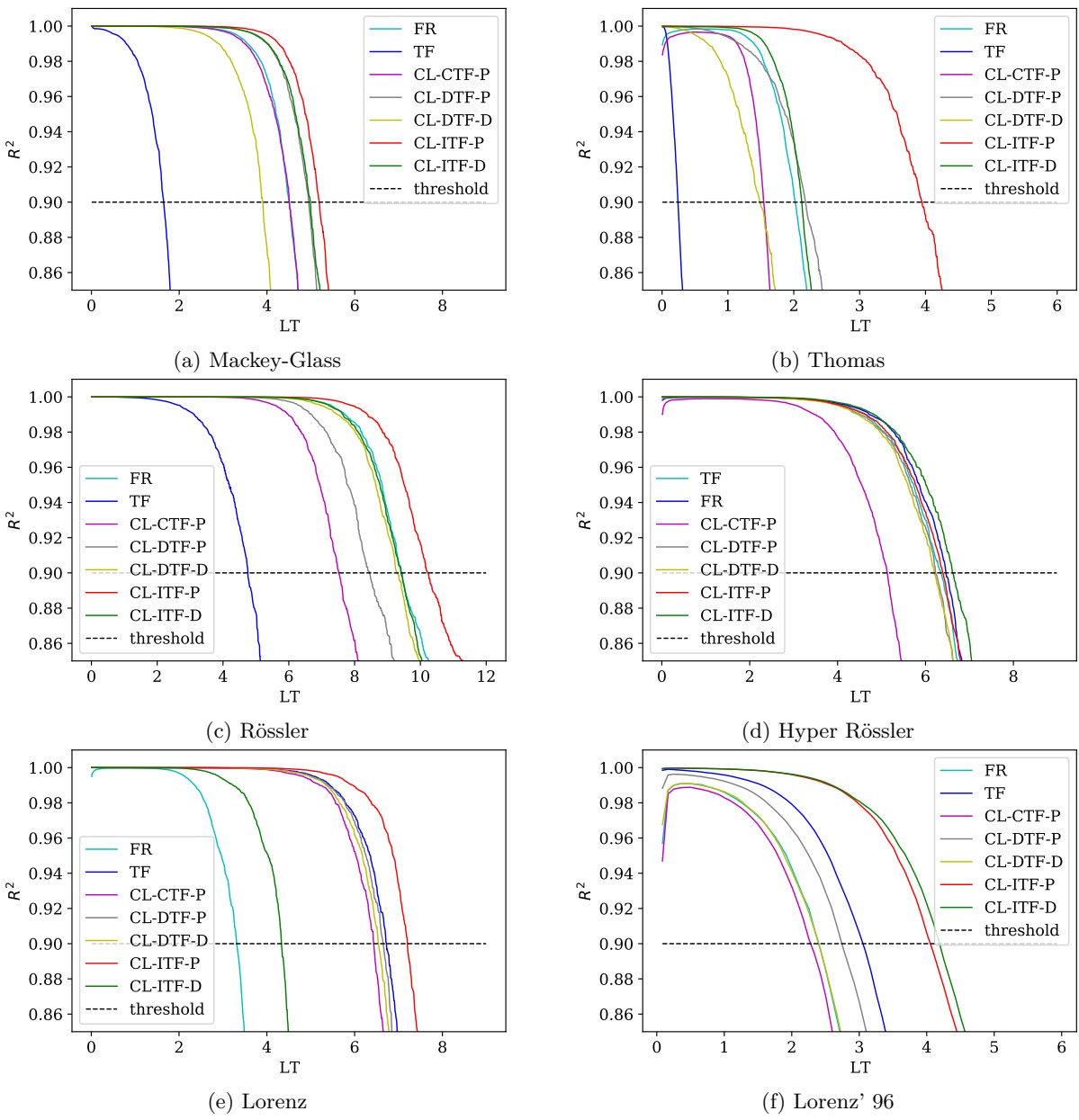

Figure 8: $R^2$ score over multiple LTs for the six studied datasets using eight different training strategies

Table V: Results of STF with those of the baseline and the CL-ITF-x strategies

| System | Strategy | Epochs | NRMSE ↓ | Rel. impr. ↑ |
|---|---|---|---|---|
| Mackey-Glass (0.006) | FR | 4 713 | 0.00391 | – |
| | TF | 44 | 0.09535 | – |
| | CL-ITF-P | 1 566 | 0.00104 | 73.40% |
| | CL-ITF-D | 1 808 | **0.00211** | 46.03% |
| | STF | 5 517 | 0.00254 | 35.04% |
| Thomas (0.055) | FR | 427 | 0.03416 | – |
| | TF | 163 | 0.34545 | – |
| | CL-ITF-P | 677 | **0.00930** | 72.78% |
| | CL-ITF-D | 649 | 0.01819 | 46.75% |
| | STF | 432 | 0.03655 | −7.00% |
| Rössler (0.069) | FR | 3 863 | 0.00098 | – |
| | TF | 500 | 0.00743 | – |
| | CL-ITF-P | 4 523 | **0.00019** | 80.61% |
| | CL-ITF-D | 7 267 | 0.00027 | 72.24% |
| | STF | 4 796 | 0.00065 | 33.67% |
| Hyper-Rössler (0.14) | FR | 6 461 | 0.00599 | – |
| | TF | 2 788 | 0.00435 | – |
| | CL-ITF-P | 2 802 | 0.00366 | 15.86% |
| | CL-ITF-D | 3 317 | 0.00326 | 25.06% |
| | STF | 2 645 | **0.00321** | 26.21% |
| Lorenz (0.905) | FR | 9 18 | 0.01209 | – |
| | TF | 4 67 | 0.00152 | – |
| | CL-ITF-P | 1 137 | **0.00060** | 60.53% |
| | CL-ITF-D | 5 78 | 0.00135 | 11.18% |
| | STF | 1 853 | 0.00511 | −236.18% |
| Lorenz'96 (1.67) | FR | 8 125 | 0.07273 | – |
| | TF | 4 175 | 0.03805 | – |
| | CL-ITF-P | 3 886 | 0.01680 | 55.85% |
| | CL-ITF-D | 3 379 | **0.01628** | 57.21% |
| | STF | 1 478 | 0.09030 | −137.32% |

using knowledge about the processed data, CL helps the model to find more consistent minima in general disregarding the degree of chaos. We hypothesize that the GRU is in many cases able to keep the risk of exploding gradients low due to its gating mechanism and thus prevents STF to really show its full strength here.

For further investigation on CL for non-chaotic systems and to enrich our experiments, we conduct additional experiments that include the application of the baseline strategies TF and FR together with CL-ITF-P on a periodic system and a measured real-world dataset. We use CL-ITF-P since it provides the most consistent relative improvements in the essential experiments. As periodic system, we study the Thomas attractor (Thomas, 1999) with parameter $b = 0.32899$ which ensures a periodic behavior. Extending our evaluation to

empirical data, we selected a time series used in the Santa Fe Institute competition (Weigend & Gershenfeld, 1993)[2].

Table VI: Comparing baseline strategies and CL-ITF-P on periodic Thomas and measured Santa Fe laser dataset

|  | Strategy | Ł | Epochs | NRMSE ↓ | Rel. impr. ↑ |
|---|---|---|---|---|---|
| Per. Thomas | FR | – | 542 | 0.00057 | – |
|  | TF | – | 542 | 0.00107 | – |
|  | CL-ITF-P | 8 000 | 794 | 0.00033 | 42.11% |
|  | CL-ITF-D | 125 | 326 | 0.00045 | 21.05% |
| Santa Fe | FR | – | 500 | 0.02170 | – |
|  | TF | – | 22 | 0.04793 | – |
|  | CL-ITF-P | 32 000 | 469 | 0.02042 | 5.90% |
|  | CL-ITF-D | 4 000 | 536 | 0.02232 | −2.86% |

The results in Table VI support our assumption that CL-ITF-x strategies are not only applicable for chaotic data originating from known dynamical systems, but also for dynamical systems with periodic behaviour achieving relative improvements of 21.05 – 42.11%. Regarding the Santa Fe dataset we observe less impact by our strategies. Only having an improvement by 5.90% for CL-ITF-P and a worsening by 2.86% for CL-ITF-D on the empirical real-world data.

We also conducted experiments for other RNN architectures, i.e., we selected a vanilla RNN and a LSTM, in the same encoder-decoder setup as applied for the previous experiments with the GRU architecture. In Tables VII and VIII, we compare the NRMSE and relative improvement of these architectures on the four chaotic datasets from the exploratory experiments (cp. Tab. III). We compare TF, FR and previously best performing training strategies CL-ITF-P and CL-ITF-D (cp. Tab. IV). Except for one case, we observe that both CL strategies outperform the respective best performing baseline strategy on the vanilla RNN as well as the LSTM architecture. The only exception is a vanilla RNN trained to forecast the Lorenz'96 system using CL-ITF-P. This setup suffers a performance decrease by 83.45% while the NRMSE in all other experiments improves by 37.83 – 75.28% RNN and 26.04 – 69.75% LSTM respectively when using the CL strategies.

## 6 Discussion

**Baseline teaching strategies (RQ1).** Considering the baseline teaching strategies FR and TF, we observe that per dataset one of the strategies performs substantially better than the other. We also observe that for the upper, based on their LLE, less chaotic datasets in Table IV FR performs better, while for the lower more chaotic datasets TF yields the better performing model. However, a larger study with more datasets would be required to justify this claim. Our takeaway is that none of the methods can universally recommended again motivating curriculum learning strategies.

**Curriculum learning strategies (RQ2).** Among the curriculum learning strategies, we observe that blending FR with a constant ratio of TF, i.e., CL-CTF-P, almost consistently yields worse results than the best performing baseline strategy and we therefore consider the strategy not relevant. The decreasing curricula CL-DTF-x that start the training with a high degree of teacher forcing and then incrementally reduce it to pure FR training partly perform better than the CL-CTF-P strategy and for a few datasets even substantially better than the baseline. However, was not foreseeable when this would be the case making their application not suitable for new datasets without a lot of experimentation and tuning. This finding is especially interesting since these strategies are conceptually similar to the scheduled sampling approach proposed for NLP tasks, thereby underlining the difference between NLP and dynamical system forecasting. We also proposed and studied increasing curricula CL-ITF-x that start the training with no or a low degree

---

[2]https://github.com/tailhq/DynaML/blob/master/data/santafelaser.csv

Table VII: Forecasting performance of the vanilla RNN on the different chaotic datasets

| System | Strategy | Ł | Epochs | NRMSE ↓ | Rel. impr. ↑ |
|---|---|---|---|---|---|
| Thomas | FR | – | 51 | 0.48117 | – |
| | TF | – | 249 | 0.41274 | – |
| | CL-ITF-P | 1 000 | 266 | **0.17955** | 56.50% |
| | CL-ITF-D | 250 | 183 | 0.25659 | 37.83% |
| Rössler | FR | – | 2292 | 0.00747 | – |
| | TF | – | 1 | 0.50174 | – |
| | CL-ITF-P | 500 | 2063 | **0.00283** | 62.12% |
| | CL-ITF-D | 1000∗ | 3075∗ | 0.00319 | 57.30% |
| Lorenz | FR | – | 506 | 0.11389 | – |
| | TF | – | 572 | 0.00913 | – |
| | CL-ITF-P | 1000 | 746 | 0.00603 | 33.95% |
| | CL-ITF-D | 125 | 782 | **0.00378** | 58.60% |
| Lorenz'96 | FR | – | 1710 | 0.31002 | – |
| | TF | – | 637 | 0.57870 | – |
| | CL-ITF-P | 1000 | 505 | 0.56872 | −83.45% |
| | CL-ITF-D | 1000 | 6573 | **0.07663** | 75.28% |

Table VIII: Forecasting performance of the LSTM on the different chaotic datasets.

| System | Strategy | Ł | Epochs | NRMSE ↓ | Rel. impr. ↑ |
|---|---|---|---|---|---|
| Thomas | FR | – | 45 | 0.43698 | – |
| | TF | – | 818 | 0.05265 | – |
| | CL-ITF-P | 250 | 758 | 0.01892 | 64.06% |
| | CL-ITF-D | 125 | 859 | **0.01181** | 77.57% |
| Rössler | FR | – | 2417 | 0.00210 | – |
| | TF | – | 1650 | 0.00139 | – |
| | CL-ITF-P | 1000 | 3426 | **0.00063** | 54.68% |
| | CL-ITF-D | 250 | 2367 | 0.00085 | 38.85% |
| Lorenz | FR | – | 1154 | 0.06526 | – |
| | TF | – | 398 | 0.00169 | – |
| | CL-ITF-P | 250 | 806 | **0.00075** | 55.62% |
| | CL-ITF-D | 31 | 419 | 0.00125 | 26.04% |
| Lorenz'96 | FR | – | 4019 | 0.13010 | – |
| | TF | – | 3721 | 0.22757 | – |
| | CL-ITF-P | 500 | 9164 | **0.03935** | 69.75% |
| | CL-ITF-D | 62 | 4509 | 0.06855 | 47.31% |

of TF, which is then incrementally increased of the course of the training. We observe that these strategies consistently outperform not only the baseline strategies but all other curriculum learning strategies as well.

**Training length (RQ3).** Choosing an improper teaching strategy can result in an early convergence on a high level of generalization error, e.g., TF strategy for Mackey-Glass, Thomas, and Rössler. Models that yield better performance typically train for more iterations (cp. Tab. III and IV). However, a longer training may not necessarily yield a better performance, e.g., FR vs. TF for Lorenz. When considering

the best performing CL-ITF-x strategies compared to the best performing baseline strategy, we observe moderately increased training iterations for some datasets, i.e., Thomas, Rössler, Hyper Rössler, Lorenz, but also decreased training iterations for other datasets, i.e., Mackey Glass and Lorenz'96. To better understand whether the longer training is the true reason for the better performing CL-ITF-x models, we compared their performance when only trained for as many iterations as the baseline model and still observe superior performance over the baseline model. In conclusion, we observe that the CL-ITF-x strategies facilitate a robust training reaching a better generalizing model in a comparable training time.

**Prediction stability (RQ4).** We evaluated the generalization as NRMSE for all models trained with the different training strategies per dataset while forecasting a dataset-specific horizon of 1 LT. However, this metric reflects only an average. When we strive for higher model performance on a multi-step prediction task we often aim for a longer prediction horizon at an acceptable error. To compare prediction stability, we report the NRMSE metric separately solely computed on the last 10% of the 1LT horizon and additionally, we computed how many LTs per datasets can be predicted before falling below an $R^2$ of 0.9. We found that the CL-ITF-x strategies yielded the lowest NRMSE of the last 10% predicted values across all datasets and even more promising that these strategies facilitated the longest prediction horizon without before falling below $R^2 = 0.9$. We conclude that the CL-ITF-x strategies train models that are substantially more stable in their long-term forecasting.

**Curriculum parametrization (RQ5).** Initially, we evaluated curriculum learning strategies with a variety of different transition functions and individual start and end teacher forcing rate (cp. *exploratory* experiments). In these experiments, we observed high prediction performance of the CL-ITF-x strategies but with a diverse dataset-specific best performing curriculum meaning that the application of these strategies for new datasets would have necessitated an extensive hyper-parameter search. In a second set of *essential* experiments, we therefore explored whether we could identify curricula with less parametrization and a similar performance. We found those by using a linear transition that is solely configured by a single parameter Ł that determines the pace with which the teacher forcing increases or decreases over the course of the training. We found that these curricula were not only comparable to the previous transition functions and their parametrization but performed better for all four datasets that we evaluated in both experimental sets and yielded the best performing model across all six datasets in the second experiment.

**Iteration scale curriculum (RQ6).** Having the CL-ITF-x strategies outperforming the CL-DTF-x strategies leads to rethinking the hitherto common intuition of supporting the early phases of training by TF and moving towards FR in the later stages of training. Rather, we hypothesize that this lures the model into regions of only seemingly stable minima, resulting in a premature termination of the training. The difference between the two CL-ITF-x strategies is how the prescribed amount of teacher forcing is distributed across the prediction steps of one training iteration (aka epoch). While the CL-ITF-D strategy distributes them as one cohesive sequence, the CL-ITF-P strategy distributes them randomly across the training sequence. We found that in the *essential* experiments with the linear transition the CL-ITF-P strategy performed overall best for four of the six datasets and would have also be a good choice with a substantial gain over the best performing baseline training strategy for the other two datasets. In conclusion, we observe that the CL-ITF-P strategy trains models that yield 16–81% higher performance than a conventional training with FR or TF. Apart from that, the *essential* results do not lead to a clear conclusion whether to use CL-ITF-P or CL-ITF-D in a given case. The above-mentioned most obvious difference in the distribution of TF steps firstly may lead to a more coherent backpropagation in the deterministic variant, but it also results in a different behavior regarding maximum number of consecutive FR steps (TF-gap) for a given $\epsilon$. Having the same curriculum function applied for CL-ITF-P and CL-ITF-D, therefore makes the TF-gap decrease much faster in the early training stage for the probabilistic variant. Further, it changes the TF-gap in a logarithmic rather than a linear fashion as for CL-ITF-D. This difference cannot be compensated by parametrizing the curriculum length demonstrating the need for the two strategies. Plus, this only affects the mean TF-gap produced by TF-ITF-P, which has a variance of $\frac{1-\epsilon}{\epsilon^2}$ due to its geometric distribution. Therefore the TF-gap also varies a lot in early training stage.

**Limitations of this work.** Our observations allow us to draw conclusions regarding appropriate curricula for the training of seq-to-seq RNN on continuous time series data. Where in our study, this data origins from a possibly unknown dynamical system that may impose chaotic behavior. However, we acknowledge that

more research is necessary to clarify the currently uncertain points. First, regarding the question why an increasing curriculum learning improves the results throughout all studied datasets. Leading to the question, what determines a proper curriculum and its parametrization. To answer these questions a closer look at the weights and behavior of the model gradients during training, the statistics of the gradient of the processed time series and the used sampling rate will be required at least. We hypothesize that this will enable us to guide the determination of the curricula's hyper-parameters and potentially allow to determine them from the characteristics of a dataset. This also includes a more thorough investigation on empirical real-world data improving on the early and inconclusive results on the Santa Fe dataset (cp. Tab. VI).

## 7 Conclusions

While training encoder-decoder RNNs to forecast time series data, strategies like teacher forcing or the from NLP tasks originating scheduled sampling are used to reduce the discrepancy between training and inference mode, i.e., the *exposure bias*. We run an extensive series of experiments using eight chaotic dynamical systems as benchmark and observed that neither of those is well suited to consistently yielding well-performing models not impacted by the *exposure bias* problem. Further, we proposed two novel curriculum learning strategies and found that those yield models that consistently outperform the best performing baseline model by a wide margin of 15–80% NRMSE in a multi-step prediction scenario over one Lyapunov time. We found that these models are more robust in their prediction allowing to forecast longer horizon with a higher performance. We found it sufficient to parametrize the strategy with a single additional parameter adopting the pace of the curriculum.

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

# A    Appendix

## A.1    Training and Validation Loss Curves

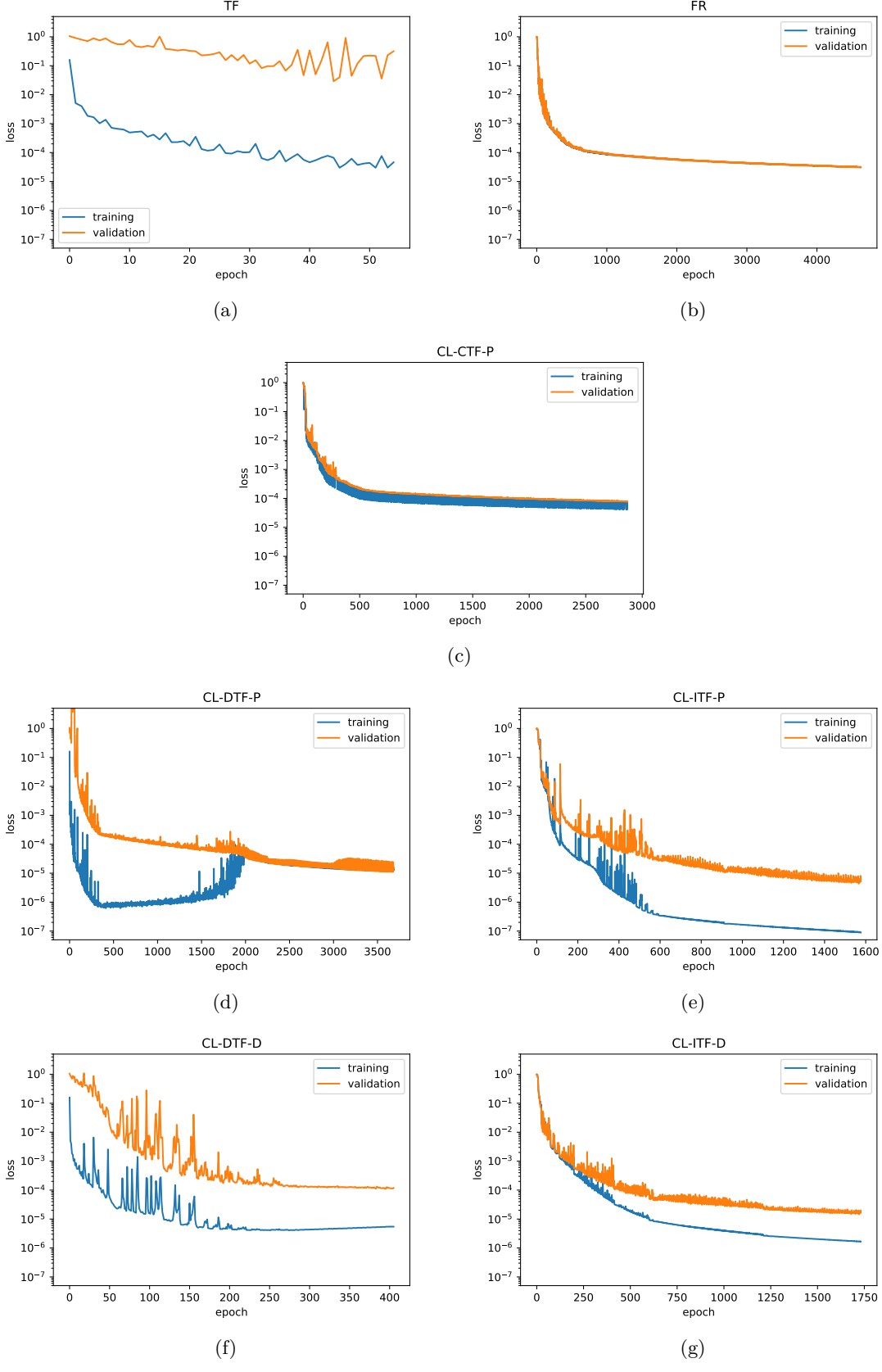

Figure 9: Training and validation loss for Mackey-Glass

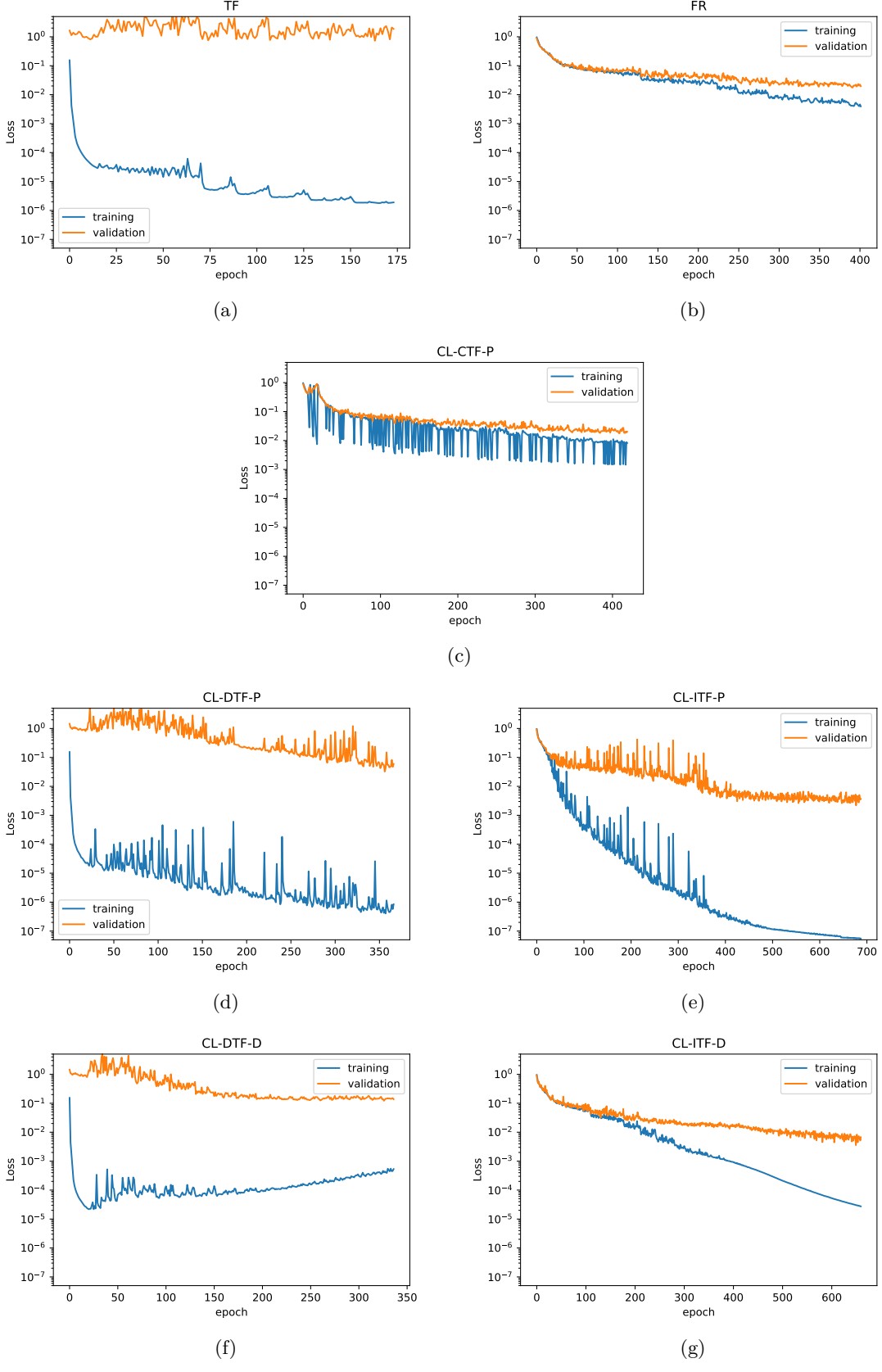

Figure 10: Training and validation loss for Thomas

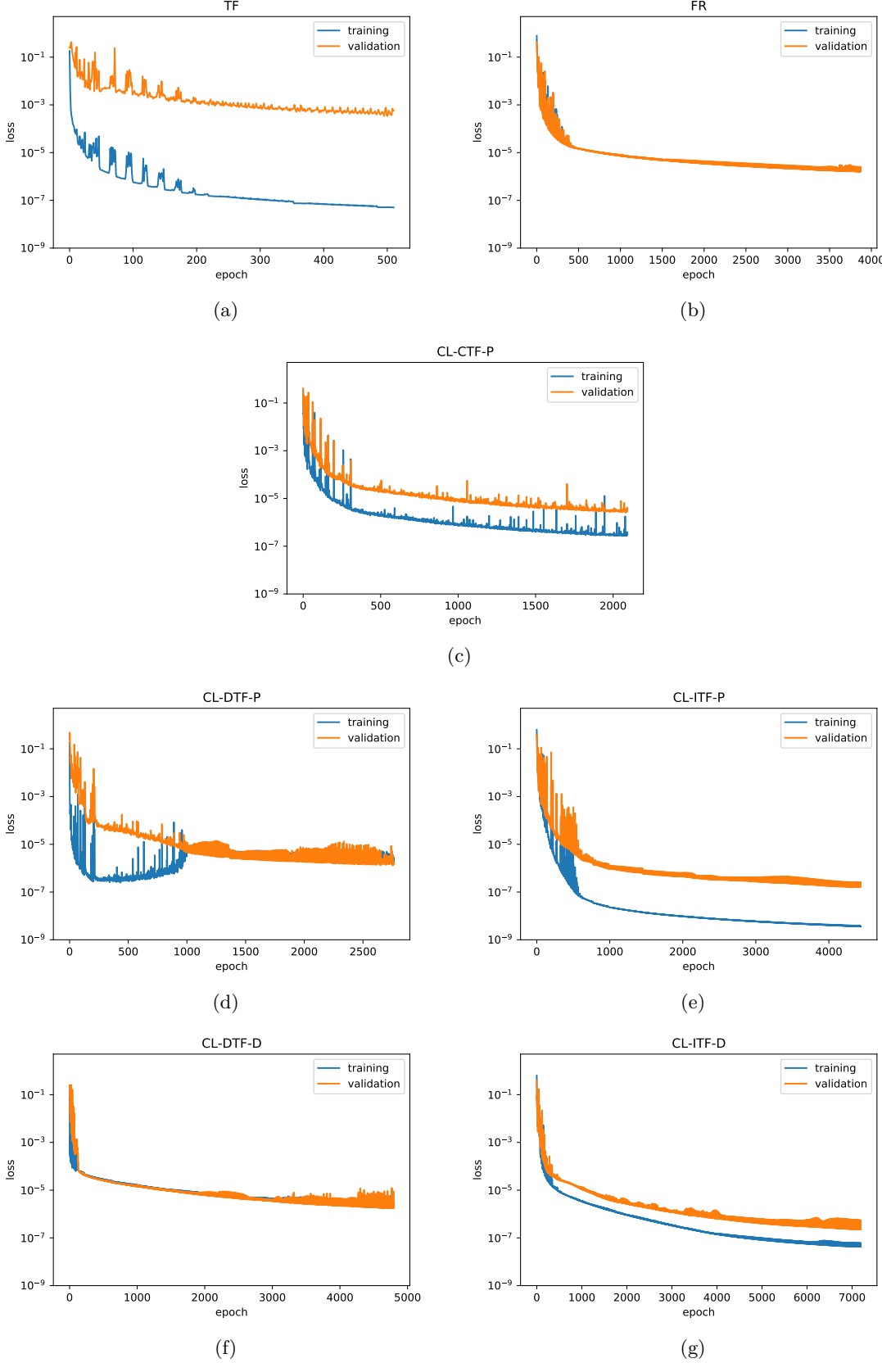

Figure 11: Training and validation loss for Rössler

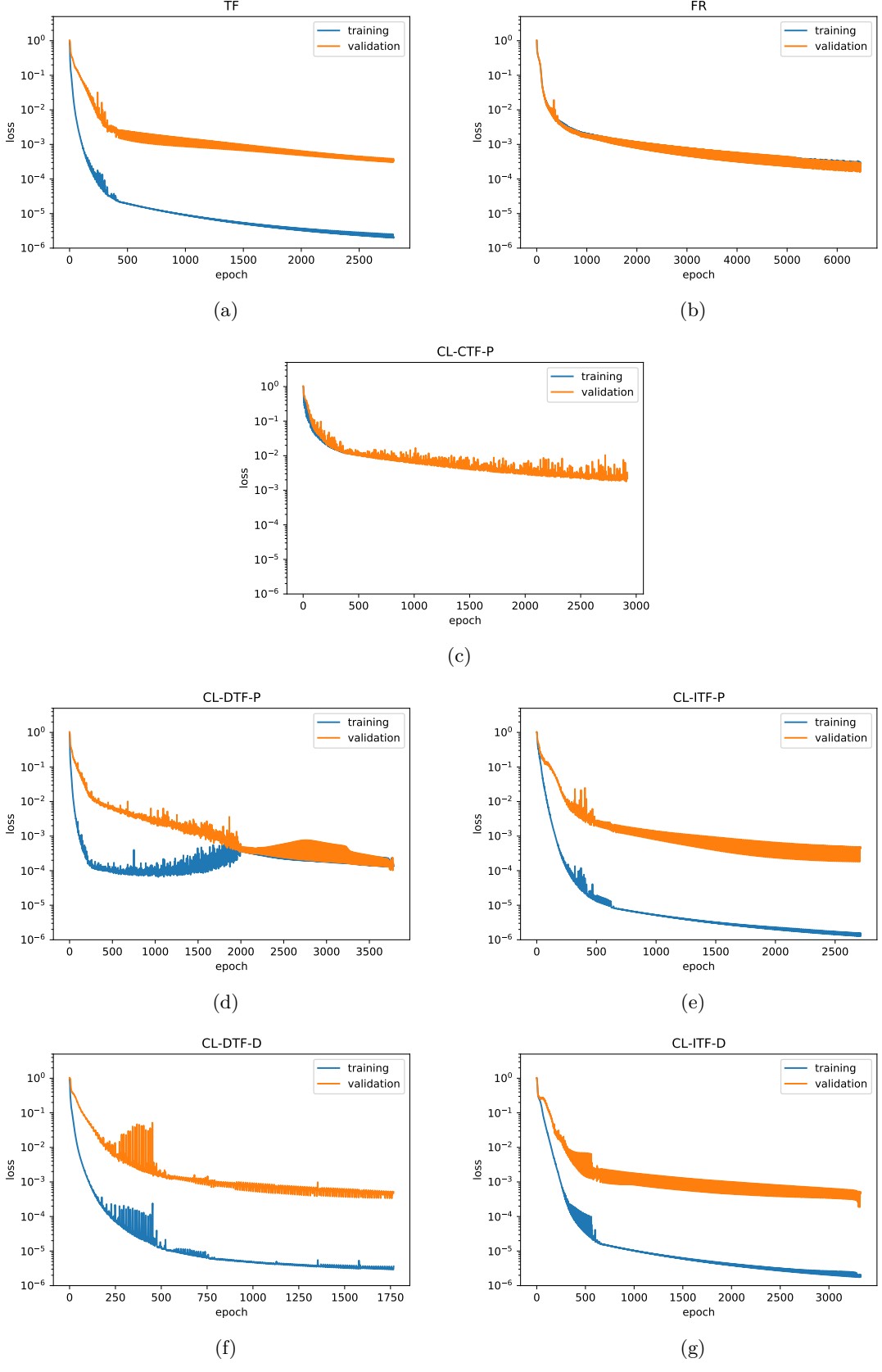

Figure 12: Training and validation loss for Hyper-Rössler

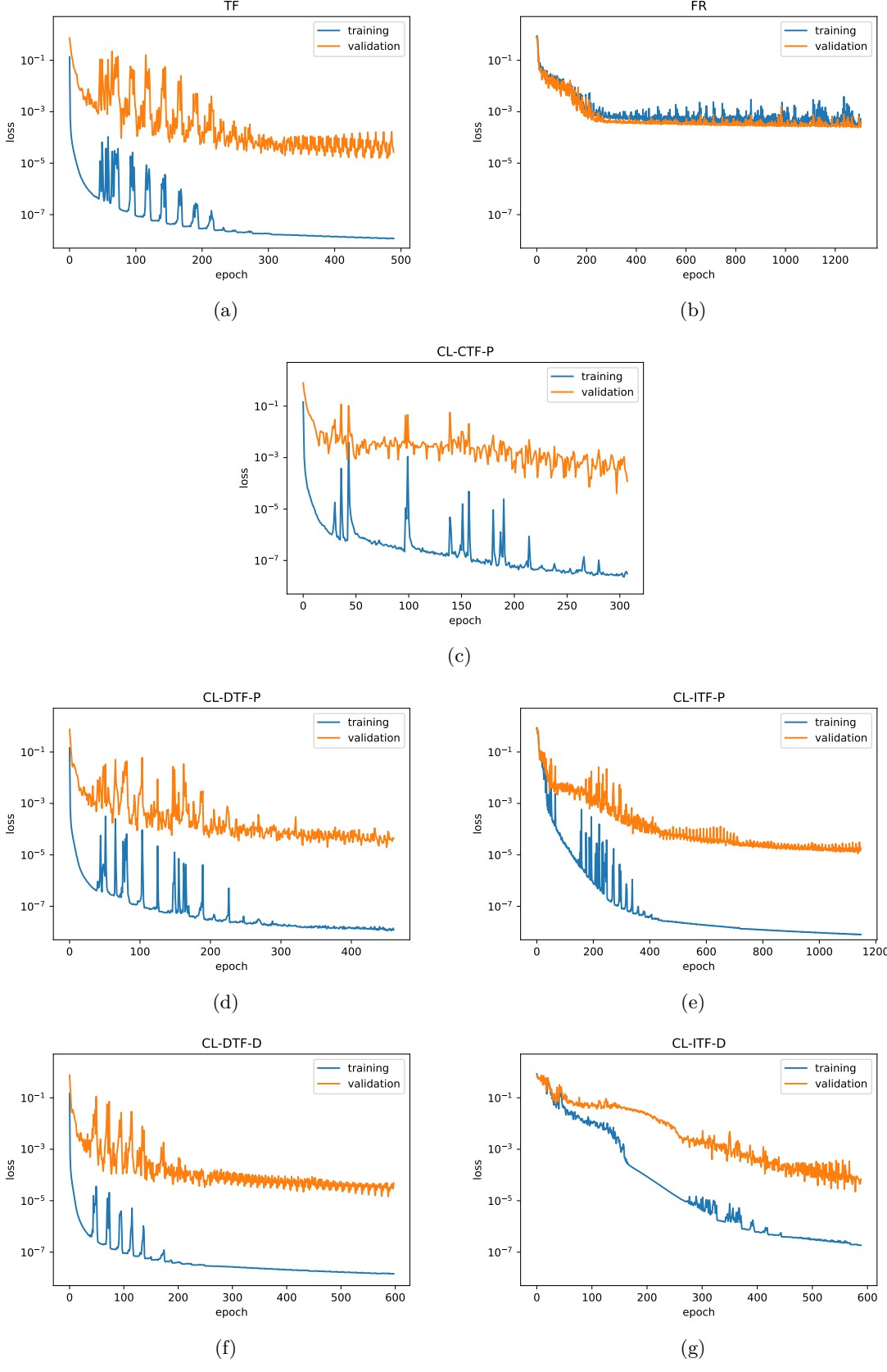

Figure 13: Training and validation loss for Lorenz

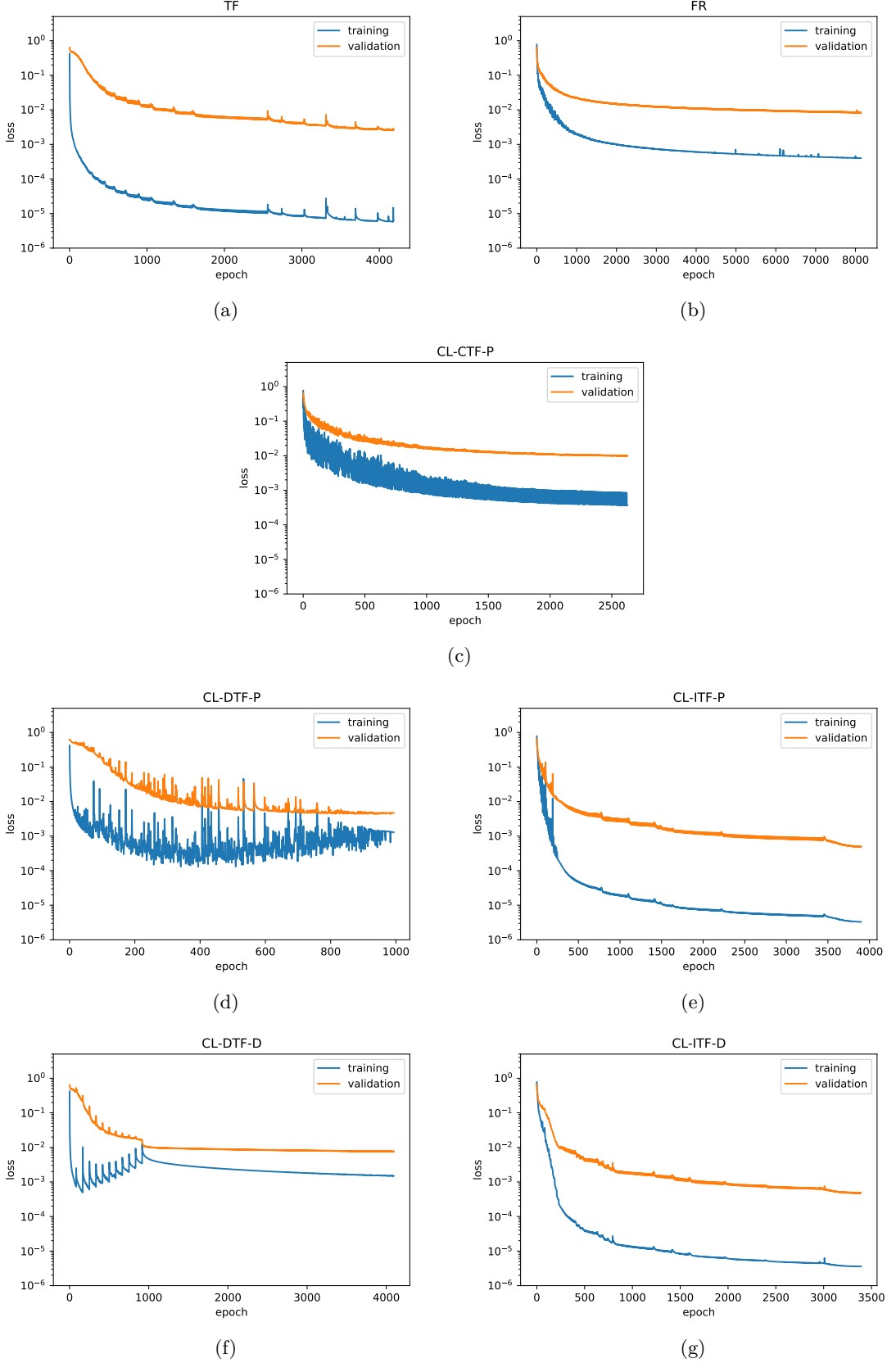

Figure 14: Training and validation loss for Lorenz'96

