# OpenReview forum: "Flipped Classroom: Effective Teaching for Chaotic Time Series Forecasting"
_TMLR — Rejected by TMLR_

### Review · Reviewer_z1SF · 2022-04-19

**Summary Of Contributions:**

Remark: I'm not familiar with the time series domain. All my comments are based on my expertise in natural language processing and deep learning. It is possible that my comments are biased.

This article studies the exposure bias in chaotic time series forecasting. Exposure bias, which is commonly available in sequence prediction for many NLP tasks, has been widely explored in literature. Scheduled sampling (Bengio et al., 2015), which is one of most effective solutions to migrate this issue, has been adopted by authors.
Their main contribution is two folds:
1) demonstrate that exposure bias is also a big issue in chaotic time series;
2) provide extensive empirical experiments of these training strategies on chaotic time series.

**Broader Impact Concerns:**

No.

**Requested Changes:**

The proposed training strategies require more training steps or are longer to converge. It may be more informative if it could provide training curves and have more analysis.

In machine translation, if we increase the size of the training set, the gap between scheduled sampling and teacher force becomes smaller. It would be interesting to how the behaviors of the proposed curricula in different training sizes.


**Strengths And Weaknesses:**

Strengths:

1. demonstrate exposure bias is the issue in chaotic time series.

2. provide extensive experiments to study different training strategies showing that the probabilistic iteration scale curriculum works better.


Weaknesses:

1. Limited novelty. These training strategies are highly similar to ones from scheduled sampling (Bengio et al., 2015).

2. Introduced more free hyper-parameter needed to tune.

---

> ### Author Response · Authors · 2022-05-20
> **Response to Request 1.1**
>
> R1.1: The proposed training strategies require more training steps or are longer to converge. It may be more informative if it could provide training curves and have more analysis.
>
> A1.1: Thanks for this comment. We added loss curves for the essential experiments to the supplementary material and added the following discussion to the end of Section 5.5:
> “We provide plots of the training and validation loss curves of the final parametrization per strategy and dataset in Appendix A.1. Based on these curves, we observe for ITF in contrast to DTF strategies that the training loss tends to move away from the validation loss faster. This is explainable by the fact that with increasing training time ITF strategies deliver an increasing amount of TF inputs counteracting the accumulation of error along the forecasted sequence and therefore further reducing training loss. For DTF strategies we observe an opposing behavior. Regarding training iterations, we observe that ITF strategies typically train for a larger number of epochs. Since the termination of the training is determined by the early stopping criterion this shows that ITF facilitates a longer and typically more successful training process compared to DTF and baseline strategies.”

---

> ### Author Response · Authors · 2022-05-20
> **Response to Request 1.2**
>
> R1.2: In machine translation, if we increase the size of the training set, the gap between scheduled sampling and teacher force becomes smaller. It would be interesting to how the behaviors of the proposed curricula in different training sizes.
>
> A1.2: We run additional experiments in this direction and compared the performance of TF and FR as now described in Section 4.1 as follows:
> “Schmidt (2019) describes the exposure bias in natural language generation as a lack of generalization. Following this argumentation motivates an analysis of training with FR and TF strategies when applied to forecasting dynamical systems with different amounts of available training data. Figure 5 shows the NRMSE when forecasting the Thomas attractor using different dataset sizes and reveals that increasing the dataset size yields generally improved model performance for TF as well as FR, while their relative difference is maintained.”

---

### Review · Reviewer_GSBp · 2022-04-22

**Summary Of Contributions:**

The authors compare different teacher forcing training schemes on time series from chaotic systems, which exhibit different degrees of chaos as assessed by the Lyapunov spectrum. The experiments are based on a GRU encoder-decoder architecture. Prediction errors for different ahead prediction time steps, immediate forecasts or relative to the Lyapunov time, are used for evaluation. Different training protocols seem optimal in different situations, but generally 1) adaptive curriculum schemes appear to work better than fixed ‘free running’ or teacher forcing schemes, 2) curriculum schemes with forcing increasing over time come out best most often.

**Broader Impact Concerns:**

None.

**Requested Changes:**

Addressing points 1-3 above I think is critical.
Specifically:
- A wider range of architectures (1), including SOTA architectures and those that have been specifically designed with chaotic dynamical systems in mind (3), should be tested.
- More theoretical analysis of why and when which curriculum scheme works best should be provided.


**Strengths And Weaknesses:**

Strengths:
The study provides a systematic comparison of different training schedules, suggesting that curriculum schemes with increasing forcing may be beneficial most commonly. The study also puts a new emphasis on finding optimal training schemes (as has been advocated recently by some authors).

Weaknesses:
First, the study does not really provide any novel methodological or conceptual developments. The types of training protocols explored in essence, more or less, have been around before. This in itself might not be a major weakness, but in my mind it raises the bar for other contributions of the paper. Knowing that certain training schemes could strongly improve performance on challenging tasks may still be very helpful for the community, but in order to draw any stronger conclusions here

1)  in my mind the results remain too heterogeneous and limited in scope. For instance, solely GRUs were tested, not any of the more recent and state-of-the-art RNNs like Lipschitz RNN, coRNN, regularized PLRNN,  antisymmetric RNN, LEM, Neural ODE etc. If one wants to take home anything of broader relevance and applicability from this study, I think a variety of RNNs needs to be tested, including state-of-the-art ones, not just older designs, to see if the results are more generic.

2) at least some theoretical analysis or insight is required. As it stands, the study remains purely exploratory and offers no theoretical guidance on how the examined curriculum schemes may work or why and when one or another scheme may be beneficial. Loss curves and gradients could be analyzed in more detail, some derivations on how the training schemes will affect the learning process seems feasible. Intuitively, one would expect that strong forcing is needed initially when the  RNN is still far from a good solution, and should be relaxed later on as the RNN is trained to capture longer and longer horizons. Why is, apparently, the opposite the case? This begs for more theoretical insight.

3) a better fit between network architecture used and the specific problem addressed is necessary in my mind. The authors basically copied a design developed for sequence-to-sequence tasks in NLP and use it for forecasting chaotic dynamical systems. But there is a huge (not at all covered) literature on machine learning for dynamical systems identification and prediction which deals at length with these problems (to give just a few pointers: https://arxiv.org/abs/1904.02107, https://arxiv.org/abs/1802.07486, https://arxiv.org/abs/1710.07313, https://arxiv.org/abs/2110.07238,  https://arxiv.org/abs/1712.09707, https://arxiv.org/abs/1910.03471, https://arxiv.org/abs/2010.08895, https://arxiv.org/abs/2106.06898, https://arxiv.org/abs/2110.05266). In my mind this would be the proper set of references and benchmark models for testing improvements on predicting chaotic systems (of note, some of these also explicitly discuss teacher forcing).

Minor technical note: A positive Lyapunov exponent is not a sufficient condition for chaos (see some common textbooks on this topic, e.g. the behavior must also be aperiodic in the asymptotic limit).

---

> ### Author Response · Authors · 2022-05-20
> **Response to Request 2.1 (1/3)**
>
> R2.1: in my mind the results remain too heterogeneous and limited in scope. For instance, solely GRUs were tested, not any of the more recent and state-of-the-art RNNs like Lipschitz RNN, coRNN, regularized PLRNN, antisymmetric RNN, LEM, Neural ODE etc. If one wants to take home anything of broader relevance and applicability from this study, I think a variety of RNNs needs to be tested, including state-of-the-art ones, not just older designs, to see if the results are more generic.
>
> A2.1.a: We chose this common, but not specifically to the forecasting of dynamical systems adapted, architecture aiming for results that are more generally interpretable regarding the studied curricula. However, we agree that extending our study of CL strategies to other RNN architecture would strengthen our observations. Therefore, we performed additional experiments with a vanilla RNN and LSTM and present the results at the end of Section 4.1:
> “We also conducted experiments for other RNN architectures, i.e., we selected a vanilla RNN and a LSTM, in the same encoder-decoder setup as applied for the previous experiments with the GRU architecture. In Tables VII and VIII, we compare the NRMSE and relative improvement of these architectures on the four chaotic datasets from the exploratory experiments (cp. Tab. III). We compare TF, FR and previously best performing training strategies CL-ITF-P and CL-ITF-D (cp. Tab. IV). Except for one case, we observe that both CL strategies outperform the respective best performing baseline strategy on the vanilla RNN as well as the LSTM architecture. The only exception is a vanilla RNN trained to forecast the Lorenz’96 system using CL-ITF-P. This setup suffers a performance decrease by 83.45% while the NRMSE in all other experiments improves by 37.83 − 75.28% RNN and 26.04 − 69.75% LSTM respectively when using the CL strategies.”

---

> ### Author Response · Authors · 2022-05-20
> **Response to Request 2.1 (2/3)**
>
> A2.1.b: Regarding the more sophisticated and dynamical systems targeted RNN architectures, we emphasize that our study does not aim to demonstrate best possible performance for a given problem but rather aim to understand the actual effect of curricula-based training strategies for a challenging time series prediction task. Nevertheless, we agree that the mentioned approaches are important contributions and should be discussed in our study. We added these studies to the Introduction and Related Work sections of our paper together with a more exhaustive explanation, why we opted to study general (gated) RNN architectures:
> Introduction: “[…] Besides exposure bias mitigation, the field of forecasting and analyzing (chaotic) dynamical systems has intensively been studied with many RNN-based approaches proposed to stabilize the training process and preventing exploding gradients. Most of these studies propose architectural tweaks or even new RNN architectures considering the specifics of dynamical systems and their theory (Lusch et al., 2018; Vlachas et al., 2018; Schmidt et al., 2019; Champion et al., 2019; Chang et al., 2019; Rusch & Mishra, 2020; Erichson et al., 2020; Rusch et al., 2021; Li et al., 2021).
> Monfared et al. (2021) performed a theoretical analysis relating RNN dynamics to loss gradients and argue that this analysis is especially insightful for chaotic systems. With this in mind they suggest a kind of sparse teacher forcing (STF) inspired by the work of Williams & Zipser (1989) that uses information about the degree of chaos of the treated dynamical system. As a result, they form a training strategy that is applicable without any architectural adaptations and without further hyperparameters. Their results using a vanilla RNN, a piecewise linear recurrent neural network (PLRNN) and an LSTM for the Lorenz (Lorenz, 1963) and the Rössler (Rössler, 1976) systems show clear superiority of applying chaos dependent STF. Reservoir computing RNNs were successfully applied to chaotic system forecasting and analysis tasks. For example, Pathak et al. (2017) propose a reservoir computing approach that fits the attractor of chaotic systems and predicts their Lyapunov exponents.”

---

> ### Author Response · Authors · 2022-05-20
> **Response to Request 2.1 (3/3)**
>
> Related Work: “[…] In the context of RNNs for forecasting and analyzing dynamical systems, the majority of existing work deals with exploding and vanishing gradients as well as capturing long-term dependencies while preserving the expressiveness of the network. Various studies rely on methods from dynamical systems theory applied to RNN or propose new network architectures.
> Lusch et al. (2018) and Champion et al. (2019) use a modified autoencoder to learn appropriate eigenfunctions that the Koopman operator needs to linearize the nonlinear dynamics of the system. In another study, Vlachas et al. (2018) extend an LSTM model with a mean stochastic model to keep its state in the statistical steady state and prevent it from escaping the system’s attractor. Schmidt et al. (2019) propose a more generalized version of a PLRNN (Koppe et al., 2019) by utilizing a subset of regularized memory units that hold information much longer and can thus keep track of long-term dependencies while the remaining parts of the architecture are designated to approximate the fast scale dynamics of the underlying dynamical system. The Antisymmetric Recurrent Neural Network (AntisymmetricRNN) introduced by Chang et al. (2019) represents an RNN designed to inherit the stability properties of the underlying ordinary differential equation (ODE) ensuring trainability of the network together with its capability of keeping track of longterm dependencies. A similar approach has been proposed as Coupled Oscillatory Recurrent Neural Networks (coRNNs) (Rusch & Mishra, 2020) that are based on a secondary order ODEs modeling a coupled network of controlled forced and damped nonlinear oscillators. The authors prove precise bounds of the RNN’s state gradients and thus the ability of the coRNN being a possible solution for exploding or vanishing gradients. Erichson et al. (2020) propose the Lipschitz RNN having additional hidden-to-hidden matrices enabling the RNN to remain Lipschitz continuous. This stabilizes the network and alleviates the exploding and vanishing gradient problem. In Li et al. (2020; 2021), the authors propose the Fourier respectively the Markov neural operator that are built from multiple concatenated Fourier layers that directly work on the Fourier modes of the dynamical system. This way they retain major portion of the dynamics and forecast the future behavior of the system. Both, the incremental Recurrent Neural Network (IRNN) (Kag et al., 2019) and the time adaptive RNN (Kag & Saligrama, 2021) use additional recurrent iterations on each input to enable the model of coping different input time scales, where the later provides a time-varying function that adapts the model’s behavior to the time scale of the provided input.
> All of this shows the increasing interest in the application of machine learning (ML) models for forecasting and analyzing (chaotic) dynamical systems. To meet this trend, Gilpin (2021) recently published a fully featured collection of benchmark datasets being related to chaotic systems including their mathematical properties.
> A more general guide of training RNNs for chaotic systems is given by Monfared et al. (2021). They discuss under which conditions the chaotic behavior of the input destabilizes the RNN and thus leads to exploding gradients during training. As a solution they propose STF, where every $\tau$-th time step a true input value is provided (teacher forced) as input instead of the previous prediction.”

---

> ### Author Response · Authors · 2022-05-20
> **Response to Request 2.2 (1/3)**
>
> R2.2: at least some theoretical analysis or insight is required. As it stands, the study remains purely exploratory and offers no theoretical guidance on how the examined curriculum schemes may work or why and when one or another scheme may be beneficial. Loss curves and gradients could be analyzed in more detail, some derivations on how the training schemes will affect the learning process seems feasible. Intuitively, one would expect that strong forcing is needed initially when the RNN is still far from a good solution, and should be relaxed later on as the RNN is trained to capture longer and longer horizons. Why is, apparently, the opposite the case? This begs for more theoretical insight.
>
> A2.2.a: Thank you for this remark! We added the loss curves regarding the essential experiments to the supplementary material and provide analysis at the end of Section 5.5:
> See A1.1.

---

> ### Author Response · Authors · 2022-05-20
> **Response to Request 2.2 (2/3)**
>
> A2.2.b: Regarding the need of a more thorough investigation of the subjectively counterintuitive behavior we added the following statement to the discussion of RQ6:
> “Having the CL-ITF-x strategies outperforming the CL-DTF-x strategies leads to rethinking the hitherto common intuition of supporting the early phases of training by TF and moving towards FR in the later stages of training. Rather, we hypothesize that this lures the model into regions of only seemingly stable minima, resulting in a premature termination of the training.”

---

> ### Author Response · Authors · 2022-05-20
> **Response to Request 2.2 (3/3)**
>
> A2.2.c: We plan to investigate the above points more thoroughly in future work and point out the current limitations at the end of the discussion Section 6:
> “Our observations allow us to draw conclusions regarding appropriate curricula for the training of seq-to-seq RNN on continuous time series data. Where in our study, this data origins from a possibly unknown dynamical system that may impose chaotic behavior. However, we acknowledge that more research is necessary to clarify the currently uncertain points. First, regarding the question why an increasing curriculum learning improves the results throughout all studied datasets. Leading to the question, what determines a proper curriculum and its parametrization. To answer these questions a closer look at the weights and behavior of the model gradients during training, the statistics of the gradient of the processed time series and the used sampling rate will be required at least. We hypothesize that this will enable us to guide the determination of the curricula’s hyperparameters and potentially allow to determine them from the characteristics of a dataset. This also includes a more thorough investigation on empirical real-world data improving on the early and inconclusive results on the Santa Fe dataset (cp. Tab. VI).”

---

> ### Author Response · Authors · 2022-05-20
> **Response to Request 2.3**
>
> R2.3: a better fit between network architecture used and the specific problem addressed is necessary in my mind. The authors basically copied a design developed for sequence-to-sequence tasks in NLP and use it for forecasting chaotic dynamical systems. But there is a huge (not at all covered) literature on machine learning for dynamical systems identification and prediction which deals at length with these problems (to give just a few pointers: https://arxiv.org/abs/1904.02107, https://arxiv.org/abs/1802.07486, https://arxiv.org/abs/1710.07313, https://arxiv.org/abs/2110.07238, https://arxiv.org/abs/1712.09707, https://arxiv.org/abs/1910.03471, https://arxiv.org/abs/2010.08895, https://arxiv.org/abs/2106.06898, https://arxiv.org/abs/2110.05266). In my mind this would be the proper set of references and benchmark models for testing improvements on predicting chaotic systems (of note, some of these also explicitly discuss teacher forcing).
>
> A2.3: We included more approaches regarding dynamic/chaotic time series data including those recommended (A2.2). Furthermore, we conducted additional experiments using the sparse teacher forcing approach (https://arxiv.org/abs/2110.07238) and compared results with the winners of the CL approaches across all studied chaotic datasets:
> “Judging from the essential experiments CL-ITF-x are our winning strategies on the chaotic systems we tested. However, as mentioned in Section 3 there are many other approaches targeting (chaotic) dynamical system forecasting with adapted RNNs architectures that take theoretical insights of dynamical systems into account. STF (Monfared et al., 2021) does not require any architectural modifications but instead provides an adapted training strategy. It determines a time interval $\tau=\frac{ln 2}{LLE}$ that denotes how many FR steps are processed before the next TF value is used within one sequence. It is the strategy that we found most comparable to the CL approaches we study. Therefore, we executed another set of experiments where we used STF during the training of our encoder-decoder GRU for all chaotic systems in Table II. Since our data is sampled with different dt we have to redefine the time interval as $\tau=\frac{ln 2}{LLE \cdot dt}$
> The results (cp. Tab. V) show that STF provides improved performance compared to the best baseline for three of six datasets ranging from 26.21 – 46.75% relative improvement. For this it does require no additional hyperparameters if the systems LLE is known. It also beats the best performing CL strategy on the Hyper-Rössler dataset by a margin of 1.15%. For the rest of the datasets, the results stay behind those of the CL-ITF-x strategies showing a worse, i.e., increased, NRMSE by 7.00 – 236.18%. We assume that where STF systematically induces TF to catch chaos preventing exploding gradients before they appear using knowledge about the processed data, CL helps the model to find more consistent minima in general disregarding the degree of chaos. We hypothesize that the GRU is in many cases able to keep the risk of exploding gradients low due to its gating mechanism and thus prevents STF to really show its full strength here. […]”

---

> ### Author Response · Authors · 2022-05-20
> **Response to Request 2.4**
>
> R2.4: Minor technical note: A positive Lyapunov exponent is not a sufficient condition for chaos (see some common textbooks on this topic, e.g. the behavior must also be aperiodic in the asymptotic limit).
>
> A2.4: Thank you for this remark, we changed the text as follows:
> “A deterministic dynamical system with at least one positive Lyapunov exponent while being aperiodic in its asymptotic limit is called chaotic”

---

> ### Author Response · Authors · 2022-05-20
> **Response to Request 2.5**
>
> R2.5: More theoretical analysis of why and when which curriculum scheme works best should be provided.
>
> A2.5: We agree there is a need for more investigation to better understand the training behavior of seq-to-seq RNNs in general and especially when trained with CL strategies. However, we argue that this requires an additional study with a more theoretical scope than our current paper to be investigated properly. To highlight these points, we now discuss the limitations of our work at the end of Section 6:
> See last part of A2.2.

---

> ### Comment · Reviewer_GSBp · 2022-05-30
> **Comments on revisions**
>
> I thank the authors for their revisions. I acknowledge that the discussion of related literature has been largely broadened, that additional experiments were performed with vanilla RNNs, LSTMs, and using STF, and some loss curves were shown in the Appendix.
>
> Although I really appreciate the effort, I’m not quite sure how well the revisions get to the essence of my points:
>
> When mentioning the work on dynamical systems learning (point 3), this was not only about the fact that a relevant literature context had been ignored by the authors for training RNNs on chaotic systems. But more generally my impression is the authors approach the whole issue from the “wrong direction” (perhaps due to the initial lack of grounding in the respective field): they motivate their study by seq-to-seq architectures in NLP and exposure bias. But learning chaotic dynamics is in general not a seq-to-seq task, and the problems encountered may have nothing to do with exposure bias and distributional shifts (at least that was nowhere demonstrated in the present paper, but just speculated). This does not mean at all that there isn’t any virtue in studying these different training protocols, but the way the authors introduce the topic in the Introduction in my mind appears largely irrelevant to the later experiments (did I miss something?).
>
> This leads into my second point, the lack of theoretical background and analysis, which is kind of a thread throughout the whole paper: It often appears like the authors just tried out a lot of different things, as in a more elaborated form of hyper-parameter scanning, but neither was that connected back to the initial exposure bias problem used to motivate the study (does one protocol address mismatch between training and test distributions more efficiently than another?), nor was it analyzed in very systematic and theory-driven way. This point is also well illustrated by the loss curves packed into the Appendix: These are just single example curves from single experiments, as far as I could see. There is no precise quantification of the loose interpretations/ observations added to the main text, no statistics, no systematic or theoretically guided approach to the problem. It all remains kind of exploratory and anecdotal and I often missed some more scientific rigor.
>
> Finally, re my point 1, the authors add vanilla RNN and LSTM (still none of the state-of-the-art models, although perhaps ok), but I still leave the paper without any clear idea of how many of the findings are just more or less random, when one or the other protocol is supposed to work or fails. Although including other RNN architectures helps to potentially establish generality of some of the findings, it’s more this systematic analysis beyond ‘trying things out’ that I miss. Shall I from now on always use CL-ITF-P? If so, why, how would I know (without going through all the separate protocols again), and how would I determine the hyper-hyper-parameters of the protocol?

---

### Review · Reviewer_twbp · 2022-05-01

**Summary Of Contributions:**

This paper studies the risk of exposure bias in training sequential models with teacher-forcing vs free-running, i.e.,

(a) Teacher-forcing (TF): Each timestep uses the correct input from the teacher
(b) Free-running (FR): Each timesteps uses the model prediction from the previous step as the input.

Teacher-forcing leads to faster training convergence but risks exposure bias since the teacher always provides the correct inputs during the training stage, the model is not exposed to its predictions as input from the previous stages. Since there is no teacher availability during the inference stage, the model risks exposure bias.

Free-running removes the exposure bias during the inference stage as the model has been trained with its predictions from the past (and this is inline with the test time setup as the teacher is not available during the inference stage)


This paper's proposal is to provide a set of curriculum that decides what fraction of the input timesteps uses the teacher inputs and what fraction uses the model predictions from previous stages. In contrast to previous works, this paper proposes curriculum that switches between TF and FR on two different scales:
(i) depending on training iteration ( ex. start with TF in the initial iterations and then decay  to use  FR )
(ii) within a training iteration ( ex. initial timesteps use TF while later timesteps use FR)

Paper evaluates the proposed curriculum learning strategies on various chaotic dynamical systems.

**Requested Changes:**

Questions for Authors:
------------

1) Since the proposed curriculum strategies are general and do not rely on any properties of the underlying dynamical system, did the authors try these strategies for problems other than chaotic dynamical systems?

2) In some cases, the deterministic strategies seem to outperform the probabilistic ones. Is this an artifact of the way the training/test data was sampled from the dynamical system or is there any other fundamental difference between these systems to the ones where probabilistic ones come on top.

3) For fair evaluation as well as distinguishing the proposed work from related works, include baselines such as (i) "scheduled sampling", (ii) teacher-student training, (iii) attention forcing.

4) It would greatly increase the impact of the proposal if the authors include experiments on real-world datasets which may or may not come from chaotic dynamics, since the proposed curriculum is generic and should be broadly applicable.


Missing Related Works:
---------
Many RNNs that are based on dyanimcal systems are missing and may be these do not have such an exposure bias issue.

For ex.
- Antisymmetric RNN : https://openreview.net/forum?id=ryxepo0cFX
- Incremental RNN : https://openreview.net/forum?id=HylpqA4FwS
- CoRNN : https://arxiv.org/pdf/2010.00951.pdf
- Time Adaptive RNNs: https://openaccess.thecvf.com/content/CVPR2021/papers/Kag_Time_Adaptive_Recurrent_Neural_Network_CVPR_2021_paper.pdf


Nit-Picks:
--------
Page-7 top para: "(cp. blue and orange line in the figure)" There is no orange line in the figure.

In main results table (II, III, IV), it would help the reader if you indicate which System is more chaotic as mentioned in the Discussion Section (Baseline teaching strategies).

**Strengths And Weaknesses:**

Strengths:
-----------

1) Simplicity of the proposed curriculum specially the probabilistic strategies that allow switching between teacher forcing or free running.

2) Extensive empirical evaluation on different chaotic dynamical systems.


Weaknesses:
-----------

1) Lack of real-world datasets for experimental evaluation. Simulations based on chaotic dynamical systems indeed provided insights into exposure problem.


2) Missing evaluations on the existing baselines mentioned in related works such as teacher-student training, attention forcing, etc. Its unclear if these learning methods fail to overcome the exposure bias in chaotic dynamical systems. Since the proposed curriculum does not rely on any properties of the dynamical system.

3) Even the closely related "scheduled sampling" (by Bengio et al. 2015) is missing in the evaluation. Although the paper discusses such related work, it does not distinguishes itself from these strategies. Without any empirical evaluation its unclear how to judge the novelty of the proposed curriculum, since scheduled sampling also starts off with TF and eventually settles on FR during the RNN training.

---

> ### Author Response · Authors · 2022-05-20
> **Response to Request 3.1 (1/2)**
>
> R3.1: Since the proposed curriculum strategies are general and do not rely on any properties of the underlying dynamical system, did the authors try these strategies for problems other than chaotic dynamical systems?
>
> A3.1.a: That’s correct, thanks for the remark. We added an additional experiment comparing the best performer from the essential experiments with the baseline strategies. We put the results in an extra subsection 5.7:
> “[…] For further investigation on CL for non-chaotic systems and to enrich our experiments, we conduct additional experiments that include the application of the baseline strategies TF and FR together with CL-ITF-P on a periodic system and a measured real-world dataset. We use CL-ITF-P since it provides the most consistent relative improvements in the essential experiments. As periodic system, we study the Thomas attractor (Thomas, 1999) with parameter b = 0.32899 which ensures a periodic behavior. Extending our evaluation to empirical data, we selected a time series used in the Santa Fe Institute competition (Weigend & Gershenfeld, 1993).
> The results in Table VI support our assumption that CL-ITF-x strategies are not only applicable for chaotic data originating from known dynamical systems, but also for dynamical systems with periodic behavior achieving relative improvements of 21.05 – 42.11%. Regarding the Santa Fe dataset we observe less impact by our strategies. Only having an improvement by 5.90% for CL-ITF-P and a worsening by 2.86% for CL-ITF-D on the empirical real-world data.

---

> ### Author Response · Authors · 2022-05-20
> **Response to Request 3.1 (2/2)**
>
> A3.1.b: We also added a remark at the end of the discussion (Sec. 6):
> See last part of A2.2.

---

> ### Author Response · Authors · 2022-05-20
> **Response to Request 3.2**
>
> R3.2: In some cases, the deterministic strategies seem to outperform the probabilistic ones. Is this an artifact of the way the training/test data was sampled from the dynamical system or is there any other fundamental difference between these systems to the ones where probabilistic ones come on top.
>
> A3.2: Even though we did not yet identify individual properties of datasets explaining which CL-ITF-x strategy works better or worse for them, we refer to some intriguing points in the discussion paragraph regarding RQ6:
> “[…] Apart from that, the essential results do not lead to a clear conclusion whether to use CL-ITF-P or CL-ITF-D in a given case. The above-mentioned most obvious difference in the distribution of TF steps firstly may lead to a more coherent backpropagation in the deterministic variant, but it also results in a different behavior regarding maximum number of consecutive FR steps (TF-gap) for a given $\epsilon$. Having the same curriculum function applied for CL-ITF-P and CL-ITF-D, therefore, makes the TF-gap decrease much faster in the early training stage for the probabilistic variant. Further, it changes the TF-gap in a logarithmic rather than a linear fashion as for CL-ITF-D. This difference cannot be compensated by parametrizing the curriculum length demonstrating the need for the two strategies. Plus, this only affects the mean TF-gap produced by TF-ITF-P, which has a variance of $\frac{1-\epsilon}{\epsilon^2}$ due to its geometric distribution. Therefore the TF-gap also varies a lot in early training stage.”

---

> ### Author Response · Authors · 2022-05-20
> **Response to Request 3.3 (1/2)**
>
> R3.3: For fair evaluation as well as distinguishing the proposed work from related works, include baselines such as (i) "scheduled sampling", (ii) teacher-student training, (iii) attention forcing.
>
> A3.3.a: Scheduled sampling is already part of our evaluation. It is present in the principle of the CL-DTF-P strategy. We now point this out more clearly in Section 4.2 as follows:
> “In our naming scheme CL-DTF-P resembles the scheduled sampling approach proposed by Bengio et al. (2015). Below, we discuss the different types of curricula on training and iteration scale resulting in different ways for determining $\Phi$.”

---

> ### Author Response · Authors · 2022-05-20
> **Response to Request 3.3 (2/2)**
>
> A3.3.b: The other approaches for exposure bias mitigation, we have omitted because we wanted to keep comparison exclusive for those that do not require rigorous changes of the architecture and are thus easy to apply for several models.

---

> ### Author Response · Authors · 2022-05-20
> **Response to Request 3.4**
>
> R3.4: It would greatly increase the impact of the proposal if the authors include experiments on real-world datasets which may or may not come from chaotic dynamics, since the proposed curriculum is generic and should be broadly applicable.
>
> A3.4: We now provide an additional experiment on the measured Santa Fe Laser dataset. See also A3.1.

---

> ### Author Response · Authors · 2022-05-20
> **Response to Request 3.5**
>
> R3.5: Many RNNs that are based on dynamical systems are missing and may be these do not have such an exposure bias issue.
> For ex.
>
>   - Antisymmetric RNN : https://openreview.net/forum?id=ryxepo0cFX
>   - incremental RNN : https://openreview.net/forum?id=HylpqA4FwS
>   - coRNN : https://arxiv.org/pdf/2010.00951.pdf
>   - Time Adaptive RNNs: https://openaccess.thecvf.com/content/CVPR2021/papers/Kag_Time_Adaptive_Recurrent_Neural_Network_CVPR_2021_paper.pdf
>
> A3.5: Thank you for this remark, we totally agree that those and other dynamical systems inspired RNNs need to be included. We changed the Introduction and Related Work section accordingly as stated in A2.1.

---

> ### Author Response · Authors · 2022-05-20
> **Response to Request 3.6**
>
> R3.6: Page-7 top para: "(cp. blue and orange line in the figure)" There is no orange line in the figure.
>
> A3.6: Yes, thank you, we missed that when changing the colors.

---

> ### Author Response · Authors · 2022-05-20
> **Response to Request 3.7**
>
> R3.7: In main results table (II, III, IV), it would help the reader if you indicate which System is more chaotic as mentioned in the Discussion Section (Baseline teaching strategies).
>
> A3.7: We added the LLE in parenthesis to each dataset in the main result tables to give a more explicit hint regarding the degree of chaos.

---

### Decision · Action_Editors · 2022-06-07

**Recommendation:** Reject

**Comment:**

This paper examines various teacher forcing training schemes on time series from chaotic systems, finding that different curricula are optimal on different tasks, but that increasing the forcing over time often performs best. The reviewers offered split opinions, with some appreciating the extensive empirical evaluations and systematic analysis, and others questioning the novelty of the proposed forcing schedules and the robustness and strength of the conclusions. Overall and in light of these somewhat conflicting perspectives, this is a borderline paper.

One concern that surfaced in the review process was a lack of clear and convincing evidence supporting the main claims. In particular, the paper does not convey clear conclusions about when particular protocols are supposed to work better than others, and the robustness of the results may not be sufficiently well established.

In order to more fully meet the standards of TMLR, the paper should reduce the scope of its claims to more accurately reflect the scope and results of the empirical evaluations. These changes would include but not be limited to highlighting nuances/caveats and the restricted scope of the evaluations. Additionally, further effort should be devoted to framing the high-level conclusions to enhance the clarity of argument and presentation. As these modifications are not necessarily minor, the manuscript cannot be accepted in its current form. The authors are encouraged to resubmit an improved version for review.